# Dentate gyrus development requires a cortical hem-derived astrocytic scaffold

**Alessia Caramello, Christophe Galichet, Karine Rizzoti\*, Robin Lovell-Badge\***

Laboratory of Stem Cell Biology and Developmental Genetics, The Francis Crick Institute, London, United Kingdom

**Abstract** During embryonic development, radial glial cells give rise to neurons, then to astrocytes following the gliogenic switch. Timely regulation of the switch, operated by several transcription factors, is fundamental for allowing coordinated interactions between neurons and glia. We deleted the gene for one such factor, SOX9, early during mouse brain development and observed a significantly compromised dentate gyrus (DG). We dissected the origin of the defect, targeting embryonic *Sox9* deletion to either the DG neuronal progenitor domain or the adjacent cortical hem (CH). We identified in the latter previously uncharacterized ALDH1L1+ astrocytic progenitors, which form a fimbrial-specific glial scaffold necessary for neuronal progenitor migration toward the developing DG. Our results highlight an early crucial role of SOX9 for DG development through regulation of astroglial potential acquisition in the CH. Moreover, we illustrate how formation of a local network, amidst astrocytic and neuronal progenitors originating from adjacent domains, underlays brain morphogenesis.

**\*For correspondence:**
Karine.Rizzoti@crick.ac.uk (KR);
robin.lovell-badge@crick.ac.uk (RL-B)

**Competing interests:** The authors declare that no competing interests exist.

## Introduction

Neuroepithelial cells (NECs) are the origin of all neurons, glia and stem cells found in the CNS (*Paridaen and Huttner, 2014*). During early CNS development, NEC potential is initially restricted to a neuronal fate. But, at around E10.5 in the mouse, NECs undergo an irreversible switch, becoming radial glial cells (RGCs), which enable them to later generate astrocyte and oligodendrocyte progenitors (*Malatesta et al., 2008*). Spatio-temporal control of the gliogenic switch is crucial because it regulates emergence and abundance of each cell type, enabling local establishment of fundamental neuron-glia interactions which are necessary for achieving correct CNS cytoarchitecture and functionality (*Orduz et al., 2019*; *Lee et al., 2019*; *Nichols et al., 2018*). Amongst other roles, these interactions are essential to support the migration of differentiating neurons (*Cooper, 2013*). RGCs are known to guide migration of their neuronal progeny, while in the adult brain, astrocytes guide neuronal progenitors from the subventricular zone to the olfactory bulb (*Lois et al., 1996*; *Gengatharan et al., 2016*). However, because astrocytic commitment has been difficult to monitor in the embryo, since specific markers to distinguish these from RGCs were lacking until recently (*Molofsky et al., 2013*; *Weng et al., 2019*) (notably ALDH1L1, previously known as FDH, 10-formyl-tetrahydrofolate dehydrogenase [*Krupenko, 2009*]), a role for astrocytes to support migration in the embryo had not been established (*Nguyen et al., 2013*).

The dentate gyrus (DG) of the hippocampus, is a packed V-shaped layer of granule neurons involved in spatial memory formation and pattern separation (*Hainmueller and Bartos, 2020*), which hosts a niche of radial glia-like cells supporting adult neurogenesis (*Ghosh, 2019*). During embryonic development, DG granule neuron progenitors originate from the dentate neuroepithelium (DNE) or primary (1ry) matrix of the archicortex, corresponding to the ventricular zone above the cortical hem (CH; *Urbán and Guillemot, 2014*). From E14.5 in the mouse, pioneer intermediate progenitors (IPs), followed by neural stem cells (NSCs), at least some of which will likely form RG-like cells in the adult DG, delaminate to migrate extensively within the parenchyma, along the Dentate Migratory Stream

(DMS) or 2ry matrix (*Nelson et al., 2020*). From E16.5, this mixed cell population, which also comprises post-mitotic neurons at this stage, ultimately reaches the brain midline forming the 3ry matrix, where they distribute in the upper then lower blades of the DG, and differentiate into PROX1+ granule neurons (*Urbán and Guillemot, 2014*). This process continues until after birth, when 1ry and 2ry matrices eventually disappear, and formation of new granule neurons will exclusively rely on local adult neurogenesis (*Nicola et al., 2015*). The CH, which is adjacent to the DNE and subsequently develops into the underlying fimbria, is a fundamental hippocampal organizer (*Yoshida et al., 2006*). It gives rise to Cajal-Retzius (CR) cells, which regulate DG progenitor migration via secretion of Reelin and SDF1 (*Sibbe et al., 2009*; *Li et al., 2009*; *Wang et al., 2018*). Progenitor migration, along the DMS, follows the track of a GFAP+ glial scaffold, which stretches from both the DNE and fimbrial epithelium toward and around the forming DG (*Rickmann et al., 1987*). Although definitive proof is lacking, the shape and directionality of GFAP+ filaments within the scaffold suggest a supportive role for progenitor migration, both along the 2ry matrix and within the 3ry matrix (*Sibbe et al., 2009*; *Li et al., 2009*; *Barry et al., 2008*; *Piper et al., 2010*; *Zhou et al., 2004*; *Galichet et al., 2008*; *Frotscher et al., 2003*; *Heng et al., 2012*). Furthermore, GFAP expression suggests that the glial scaffold is formed of differentiating astrocytes, because GFAP is not expressed in RGCs in rodents (*Malatesta et al., 2008*). Deletion of the genes encoding the transcription factors NF1A/B, which regulate astrocyte gene expression (*Kang et al., 2012*), partly affects formation of the scaffold, also suggesting an astrocytic contribution (*Barry et al., 2008*; *Piper et al., 2010*; *Brunne et al., 2010*). However, direct evidence of astrocytes supporting neuronal progenitor migration during embryonic development has never been reported before. It is formally possible that these astrocytic cells, or at least some of them, could be RGCs, which are known to guide migration. However, we have no evidence for this given the absence of reliable distinguishing markers. The mechanisms explaining the formation of the scaffold also remain ambiguous: it has been suggested to have a dual origin with a proximal fimbrial part deriving from the fimbrial glioepithelium, and a distal supragranular domain originating from DNE progenitors (*Li et al., 2009*; *Barry et al., 2008*; *Heng et al., 2012*). Therefore, direct evidence of its role and its regional and cellular origin are lacking.

SOX9, an SRY-related high mobility group (HMG) box (SOX) transcription factor, starts to be expressed in NECs around E9.5, just before their transition to RGCs and neuro-to-glia switch. We and others showed that SOX9 is required for this process, both within the embryonic brain (*Scott et al., 2010*; *Hashimoto et al., 2016*) and spinal cord (*Kang et al., 2012*; *Stolt et al., 2003*), because generation of astrocytes and oligodendrocytes is affected by its deletion. The role of SOX9 during early CNS development has been analyzed, in particular by conditional deletion using *Nestin-Cre* (*Tronche et al., 1999*). However, this Cre-driver only becomes active from E10.5, after the onset of SOX9 expression (*Scott et al., 2010*). Consequently, the relatively mild effect of *Sox9* loss on astrogenesis in *Sox9*$^{fl/fl}$;*Nestin-Cre* mutants (*Kang et al., 2012*; *Stolt et al., 2003*) might also be explained by its early, albeit temporary expression.

To better understand the role of SOX9 in CNS development, including at early stages, we performed conditional deletion using *Sox1*$^{Cre/+}$, which is active from E8.5 almost exclusively in the neural tube (*Takashima et al., 2007*; *Wood and Episkopou, 1999*), and compared these with *Sox9*$^{fl/fl}$;*Nestin-Cre* mutants. In contrast with the latter model (*Tronche et al., 1999*), all *Sox9*$^{fl/fl}$;*Sox1*$^{Cre/+}$ mutant mice survived, allowing post-natal analysis. Reduced DG size was the most prominent defect in adult *Sox9*$^{fl/fl}$;*Sox1*$^{Cre/+}$ mutant brains, but not in the few surviving *Sox9*$^{fl/fl}$;*Nestin-Cre* animals, and this was already visible in newborns, suggesting an earlier developmental defect. While the emergence and differentiation of granule neuron progenitors were not affected in either *Sox9* mutant embryos, we observed that their migration within the developing DG was compromised, particularly in *Sox9*$^{fl/fl}$;*Sox1*$^{Cre/+}$ mutants. We then showed that formation of the fimbrial glial scaffold, which is likely supporting neuronal progenitor migration toward the forming DG, was delayed in *Sox9*$^{fl/fl}$;*Sox1*$^{Cre/+}$ mutants. We furthermore identified ALDH1L1+ astrocytic progenitors in the adjacent CH as the origin of the fimbrial glial scaffold. Accordingly, formation of these progenitors is significantly compromised in *Sox9*$^{fl/fl}$;*Sox1*$^{Cre/+}$ mutants, but not in their *Sox9*$^{fl/fl}$;*Nestin-Cre* counterparts, because *Nestin-Cre* is not active in the CH. Consequently, fimbrial glial scaffold and DG formation are less affected in these mutants. Exclusive deletion of *Sox9* in the CH, using *Wnt3a*$^{ir-esCre}$ (*Yoshida et al., 2006*), further confirmed that SOX9 is required for astrocyte progenitor emergence and hence fimbrial glial scaffold formation, allowing neuron progenitor migration. Ultimately,

our results highlight the crucial importance of the timely emergence of glial progenitors in the CH for the establishment of a local supporting cellular network underlying neuronal migration during DG morphogenesis, and that, through its role in acquisition of astroglial potential, SOX9 is critical for this.

## Results

### Adult DG morphology is sensitive to precise patterns of *Sox9* deletion in the archicortex

To further characterize the role of SOX9 during CNS development, we first performed an early CNS-specific conditional deletion of the gene by crossing *Sox9^fl/fl* (*Akiyama et al., 2002*) with *Sox1^Cre*, which is active from E8.5 (*Takashima et al., 2007*) prior to the onset of SOX9 expression (*Scott et al., 2010*). The birth, growth, and survival of *Sox9^fl/fl;Sox1^Cre/+* mice were not overtly affected. However, histological analyses of adult *Sox9* mutant brains revealed that the hippocampus was particularly affected (*Figure 1*, *Figure 1—figure supplement 1i,ii*). However, the DG emerges as the most affected region, and quantification of its size shows it was significantly reduced in adult *Sox9^fl/fl;Sox1^Cre/+* mice compared to controls (*Figure 1Ai,ii;B*). Furthermore, deletion of one copy of *Sox9* in *Sox9^fl/+;Sox1^Cre/+* mice does not affect DG size (*Figure 1*, *Figure 1—figure supplement 2*). DG size reduction was already evident at P2 (*Figure 1Aiv–v,C*), indicating this phenotype might arise earlier, due to the absence of SOX9 during DG embryonic development. Because the adult DG controls formation of new memories (*Hainmueller and Bartos, 2020*), we assessed memory formation abilities in adult *Sox9^fl/fl;Sox1^Cre/+* mice by performing a Novel Object Recognition Test (NORT) (*Figure 1D*). Failure to recognise a new object over a familiar one, detected as spending more time to investigate the new object, was demonstrated in *Sox9^fl/fl;Sox1^Cre/+* mice (*Figure 1E*). Because deficiency in memory formation could also be caused by reduced exploration of the arena due to anxiety-like behaviors, we performed in parallel an open-field test (*Figure 1*, *Figure 1—figure supplement 3A*). This did not reveal any significant difference between control and mutant mice (*Figure 1*, *Figure 1—figure supplement 3B,C*). Taken together, these results show that embryonic deletion of *Sox9* affects memory-forming abilities. This suggests that functionality of DG is affected, albeit we cannot exclude that defects in other regions of the mutant brains explain or exacerbate this behavioral deficiency.

*Sox9^fl/fl;Nestin-Cre* mutants (*Scott et al., 2010*) were generated in parallel to examine morphogenesis of the DG. In these, *Sox9* deletion occurs around 48 hr later than in *Sox9^fl/fl;Sox1^Cre/+* mutants, at E10.5, as SOX9 protein starts to be expressed in the developing CNS (*Scott et al., 2010*). In contrast with *Sox9^fl/fl;Sox1^Cre/+* mutants, most *Sox9^fl/fl;Nestin-Cre* animals die at birth (*Scott et al., 2010*). Therefore, activity of *Nestin-Cre* outside the CNS (*Bernal and Arranz, 2018*), in tissues where SOX9 is required, such as pancreatic islets (*Seymour et al., 2007*), heart (*Akiyama et al., 2004*), and/or kidneys (*Reginensi et al., 2011*), likely explains mortality in these mutants. We were, however, able to analyse two *Sox9^fl/fl;Nestin-Cre* animals that survived until around 3 months of age and in which loss of SOX9 was confirmed by immunostaining (*Figure 1*, *Figure 1—figure supplement 4*). In contrast with *Sox1^Cre/+* mutants, the size of the DG in *Sox9^fl/fl;Nestin-Cre* animals was unaffected both in adults (*Figure 1Aiii,B*) and P2 pups (*Figure 1Avi,C*).

The difference in timing and/or pattern of embryonic *Sox9* deletion may underlie the variation in DG defects among these two mutant strains. To test this hypothesis, we analyzed the activity of the *Sox1^Cre* and *Nestin-Cre* by lineage tracing using an *R26R^eYFP* allele and in parallel examined SOX9 expression. At E11.5, eYFP reporter expression was observed throughout the forebrain in *Sox9^fl/fl;Sox1^Cre/+;R26R^eYFP/+* embryos (*Figure 1Fiii*) where, compared to controls, SOX9/*Sox9* expression was absent (*Figure 1Fii,iv* and *Figure 1—figure supplement 5Ai–ii,B*). However, in *Sox9^fl/fl;Sox1-^Cre/+* mutant archicortices, we detected some rare *Sox9*-positive cells by in situ hybridization (arrow in *Figure 1—figure supplement 5Aviii*, see also Figure 5Gv), demonstrating that recombination of the *Sox9* allele is not quite ubiquitous. In contrast, *Nestin-Cre* is only active ventrally at E11.5, gradually progressing dorsally later in gestation (*Figure 1Fv–x*, *Figure 1—figure supplement 5Aiii,vi,ix*), as previously shown (*Vernay et al., 2005*). This implies that in *Sox9^fl/fl;Nestin-Cre* mutants, SOX9 is transiently expressed in the DNE between E12.5 and E13.5 (*Figure 1Giii,vi*). In addition, the adjacent CH presents a mosaic pattern of recombination in *Nestin-Cre* mutants (*Figure 1Gvi,ix,xii*), as

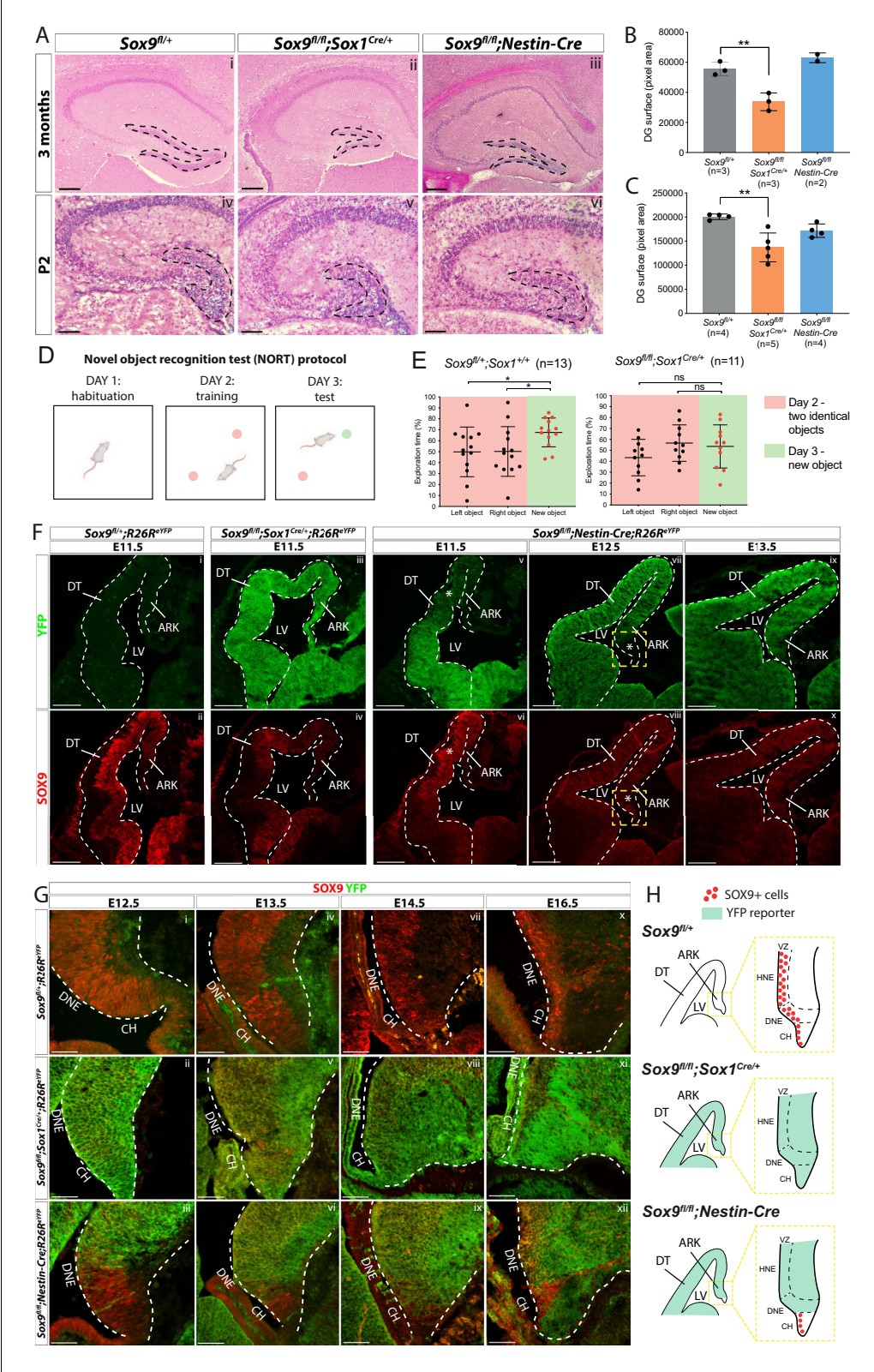

**Figure 1.** Dentate gyrus (DG) morphogenesis is differentially affected in *Sox1^{Cre}* versus *Sox9^{fl/fl}*;*Nestin-Cre* mutants. (**A**) H and E staining of 3-month-old (i-iii) and P2 (iv-vi) brain sections of *Sox9^{fl/fl}*;*Sox1^{Cre/+}* and *Sox9^{fl/fl}*;*Nestin-Cre* mutants compared to *Sox9^{fl/+}* controls. DG (outlined) appears smaller in *Sox9^{fl/fl}*;*Sox1^{Cre/+}* mutants compared to both controls and *Sox9^{fl/fl}*;*Nestin-Cre* mutants. (**B–C**) Quantification of DG surface as pixel area, in 3-month-old mice. (**B**) DG is significantly smaller in *Sox9^{fl/fl}*;*Sox1^{Cre/+}* mutants (33758 ± 5898) compared to controls (55651 ± 4492, *t* test p=0.0069) and *Sox9^{fl/fl}*;

*Figure 1 continued on next page*

*Figure 1 continued*

*Nestin-Cre* mutants (62994 ± 3243, statistical analysis for *Nestin-Cre* mutants is not possible as n < 3). The defect is already visible in P2 pups (C), when DG area is significantly smaller in *Sox9^{fl/fl};Sox1^{Cre/+}* mutants (137101 ± 29892) compared to controls (200651 ± 5683, p=0.026), but not compared to *Sox9^{fl/fl};Nestin-Cre* (171772 ± 13866, Tukey's multiple comparison test p=0.1672, ANOVA p=0.033). (D) Schematic of novel object recognition test (NORT) protocol. Pink and green circles represent familiar and new objects, respectively. (E) Quantification of exploration times spent by mice over the identical left and right object on day 2 (red boxes) and on the new object on day 3 (green boxes). *Sox9^{fl/+};Sox1^{+/+}* control mice (n = 13) spend significantly more time exploring the new object on day 3 (67.55 ± 13.16%) compared to time spend exploring the identical object on day 2 (left object: 49.83 ± 22.65%, *t* test p=0.0294; right object: 50.17 ± 22.65%, *t* test p=0.0391), indicating that they remember the objects from day 2. *Sox9^{fl/fl};Sox1^{Cre/+}* mutants (n = 11) instead do not remember the objects from day 2, because the time spent exploring the new object on day 3 (53.59 ± 19.70%) is not different from the time spent exploring objects on day 2, either on the left (43.37 ± 16.65%, *t* test p=0.2009) or right side (56.63 ± 16.65%, *t* test p=0.6839). (F–H) Immunofluorescence for YFP and SOX9 comparing, respectively, expression of *R26R^{eYFP}* reporter of Cre activity and SOX9 expression patterns in *Sox9^{fl/fl};Sox1^{Cre/+}*, *Sox9^{fl/fl};Nestin-Cre* mutants and controls, during forebrain (F) and archicortex (G) development. SOX9 remains expressed in *Nestin-Cre* mutants in both the CH and DNE (white asterisks in F). Yellow dashed square in (F) indicate area shown in (G) at higher magnification, also schematized in (H) together with *Sox1^{Cre}* and *Nestin-Cre* recombination pattern in the ARK at E13.5. Signal from SOX9 immunofluorescence in *Sox9* mutant tissue was confirmed to be background with ISH for *Sox9* (Figure S.5). LV: lateral ventricle; DT: dorsal telencephalon; ARK: archicortex; CH; cortical hem; DNE: dentate neuroepithelium; VZ: ventricular zone; HNE: hippocampal neuroepithelium; 1ry: primary matrix; 2ry: secondary matrix; 3ry: tertiary matrix. Scale bar represent 400 μm in (Ai-iii); 200 μm in (Aiv-vi) and (F); 50 μm in (G).

The online version of this article includes the following source data and figure supplement(s) for figure 1:

**Source data 1.** Quantification of dentate gyrus (DG) size in adults and P2 pups and analysis of memory formation ability during NORT behavioral test.

**Figure supplement 1.** Histological analysis of CA regions in *Sox9^{fl/fl};Sox1^{Cre/+}* adult mice.

**Figure supplement 2.** Histological analysis *Sox9^{fl/+};Sox1^{Cre/+}* adult mice adult dentate gyrus (DG).

**Figure supplement 2—source data 1.** Quantification of dentate gyrus (DG) size in *Sox9^{fl/+};Sox1^{Cre/+}* adult mouse.

**Figure supplement 3.** *Sox9^{fl/fl};Sox1^{Cre/+}* adults do not show anxiety-like behavior in open-field test.

**Figure supplement 3—source data 1.** Analysis of anxiety behavior with open-field test.

**Figure supplement 4.** Absence of SOX9 expression in a 3-month-old *Sox9^{fl/fl};Nestin-Cre* mutant brain.

**Figure supplement 5.** Qualitative and quantitative analyses of *Sox9* transcripts confirm residual *Sox9* expression in embryonic *Sox9^{fl/fl};Nestin-Cre* forebrains.

**Figure supplement 5—source data 1.** Quantification of *Sox9* expression with qPCR in E12.5 DT and ARK separately.

previously shown (*Li et al., 2009*; *Winkler et al., 2018*). Consequently, many SOX9-expressing cells can still be found at E16.5 in this region in *Sox9^{fl/fl};Nestin-Cre* mutants (*Figure 1Gxii*).

Altogether these results suggest that delayed recombination in the DNE and/or residual expression of SOX9 in the CH of *Sox9^{fl/fl};Nestin-Cre* embryos (schematized in *Figure 1H*) underlies the difference in adult DG phenotype observed between *Sox9^{fl/fl};Sox1^{Cre/+}* and *Sox9^{fl/fl};Nestin-Cre* mutants. Comparative analysis of DG development was then performed in both models to characterize the origin of the defect observed in *Sox9^{fl/fl};Sox1^{Cre/+}* mutants.

## Abnormal distribution of granule neurons and their progenitors in the developing DG of *Sox9* mutants

DG progenitors, IPs, and differentiating granule neurons were first examined by analyzing respectively the expression of the transcription factors PAX6 (*Englund et al., 2005*), TBR2 (*Hodge et al., 2012*), and PROX1 (*Lavado et al., 2010*) at different stages of embryonic (E14.5, E16.5, E18.5) and post-natal (P2) DG development in *Sox9^{fl/fl};Sox1^{Cre/+}* and *Sox9^{fl/fl};Nestin-Cre* mutants and controls (*Figure 2A,F,I* and *Figure 2—figure supplement 1A*). All three cell types were found in the same numbers at embryonic stages in both *Sox9* mutants compared to controls (*Figure 2E,G,J* and *Figure 2—figure supplement 1B,C*). However, at P2, TBR2+ IPs are reduced in *Sox9^{fl/fl};Sox1^{Cre/+}* mutants compared to controls, with 27.6% fewer cells (*Figure 2G*). PROX1+ cells also appeared to be reduced in number, although this was not statistically significant (*Figure 2J*). There were also fewer TBR2+ (*Figure 2G*) and PROX1+ (*Figure 2J*) cells were also counted in *Sox9^{fl/fl};Nestin-Cre* P2 mutants compared to controls. While these also did not reach statistical significance, these results nevertheless suggest that a subtle disruption of granule neuron progenitor formation is present in *Nestin-Cre* mutants. Analysis of cleaved Caspase-3 immunostaining showed comparable patterns of cell apoptosis between mutants and controls, suggesting *Sox9* deletion does not affect progenitor survival at this stage (*Figure 2—figure supplement 2*). Moreover, Ki67 expression and EdU labeling, revealed no difference between controls and mutants in emergence, expansion and differentiation of DG granule neuron progenitors (*Figure 2—figure supplement 3A,B,D–F,H–J*). Altogether, these

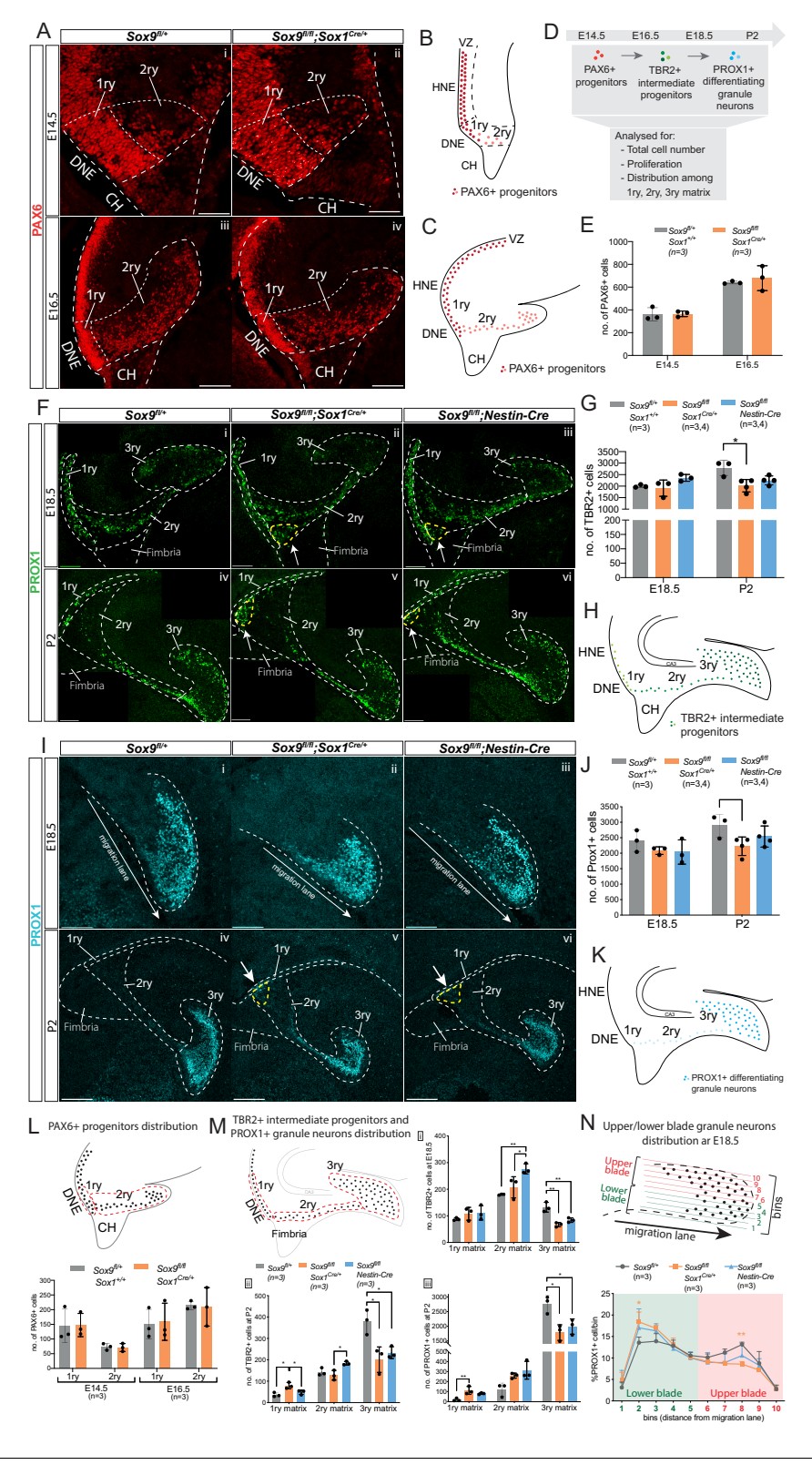

**Figure 2.** Granule neuron progenitors are generated normally but their distribution in SOX9 mutant developing DG is abnormal. (A–K) Immunofluorescence for PAX6 (A) TBR2 (F) and PROX1 (I) were performed at indicated developmental (E14.5: Ai, ii; E16.5: Aiii, iv; E18.5: Fi-iii and Ii-iii) and postnatal stages (P2: Fiv-vi and Iiv-vi) of DG development. Arrows in Fii-iii,v-vi and Iv-vi point to respectively TBR2+ and PROX1+ cells accumulating close to the ventricle (yellow dashed line). 1ry, 2ry, and 3ry matrices are delineated with white dashed lines. Localization within the developing DG of

*Figure 2 continued on next page*

*Figure 2 continued*

each cell type analyzed is schematized for PAX6 at E14.5 (B) and E16.5 (C), for TBR2 at E18.5 and P2 (H) and PROX1 for E18.5 and P2 (K), where color intensity in the illustration reflects level of markers expression. (D) shows experimental analysis, total cell number indicate sum of cells in 1ry, 2ry, and 3ry matrix. Quantification of total PAX6+ cells (E), TBR2+ cells (G) and PROX1+ cells (J) is shown at the indicated developmental and postnatal stages. Total cell number analysis shows a reduced number of TBR2+ cells at P2 (G) in *Sox9^{fl/fl};Sox1^{Cre/+}* mutants (2020.31 ± 267.74; Tukey's multiple comparison test p=0.01190, ANOVA p=0.014) compared to controls (2792 ± 331.72). The same tendency was observed for TBR2+ cells in P2 *Sox9^{fl/fl}; Nestin-Cre* mutants (G) (2249.75 ± 195.18), and for PROX1+ cells (J) in both *Sox9* mutants at P2 (*Sox9^{fl/fl};Sox1^{Cre}*: 2225.50 ± 299.24 and *Sox9^{fl/fl};Nestin-Cre*: 2538.75 ± 340.30) compared to controls (2895.33 ± 367.51). (L, M) Analysis of PAX6+, TBR2+, and PROX1+ cells distribution along the three matrices, according to the corresponding above schematics where dashed lines indicate areas considered for 1ry, 2ry, and 3ry matrix quantification (also shown in A, F, Iiv-vi). At E14.5 and E16.5 (L), the same amount of PAX6+ cells are found in the 1ry and 2ry matrix in *Sox9^{fl/fl};Sox1^{Cre/+}* mutants compared to controls. At E18.5 (Mi), more TBR2+ cells were found in the 2ry matrix of *Sox9^{fl/fl};Nestin-Cre* mutants (276.53 ± 18.96) compared to both *Sox9^{fl/fl};Sox1^{Cre/+}* mutants (207.33 ± 39.85, p=0.03660) and controls (180.07 ± 1.79, Tukey's multiple comparison test p=0.00850, ANOVA p=0.0090), while less TBR2+ cells were found in the 3ry matrix of both *Sox9^{fl/fl};Sox1^{Cre/+}* mutants (66.93 ± 7.90, p=0.0016) and *Sox9^{fl/fl};Nestin-Cre* mutants (84.00 ± 8.50, p=0.0075) compared to controls (132.53 ± 18.29, Tukey's multiple comparison test, ANOVA p=0.0017). At P2 (Mii) more TBR2+ cells are found in 1ry matrix of *Sox9^{fl/fl};Sox1^{Cre/+}* mutants (79.47+14.59), compared to controls (36.47 ± 9.87, p=0.0101) and *Sox9^{fl/fl};Nestin-Cre* mutants (48.13 ± 10.35, Tukey's multiple comparison test p=0.0399, ANOVA p=0.0106). In *Sox9^{fl/fl};Nestin-Cre* mutants, more TBR2+ cells are accumulating in the 2ry matrix (184.07 ± 8.47) compared to *Sox9^{fl/fl};Sox1^{Cre/+}* mutants (127.87 ± 22.72, Tukey's multiple comparison test p=0.0175, ANOVA p=0.0183). In both *Sox9* mutants, less TBR2+ cells are found in the 3ry matrix (*Sox9^{fl/fl};Sox1^{Cre/+}*: 201.00 ± 59.44, p=0.0119; *Sox9^{fl/fl};Nestin-Cre*: 233.73 ± 27.81, p=0.029) compared to controls (378.93 ± 57.88, Tukey's multiple comparison test, ANOVA p=0.0109). At P2, PROX1+ cells (Miii) accumulate in the 1ry matrix of *Sox9^{fl/fl};Sox1^{Cre/+}* mutants, (111.00 ± 39.89) compared to controls (17.67 ± 14.15, Tukey's multiple comparison test p=0.0088, ANOVA p=0.0100), and a significant decrease in the 3ry matrix of both *Sox9^{fl/fl};Sox1^{Cre/+}* mutants (1786.67 ± 266.25, p=0.0117) and *Sox9^{fl/fl};Nestin-Cre* mutants (1991.33±260.48, p=0.0329) is observed compared to controls (2758.33 ± 297.16, Tukey's multiple comparison test, ANOVA p=0.0112). (N) Analysis of the distribution of PROX1+ granule neurons distribution within the upper and lower blade of the forming DG at E18.5: the 3ry matrix was divided in 10 horizonal ventral to dorsal bins spanning the lower to upper blade domain. Cells were then counted within each bin. The percentage of PROX1+ cells present in each bin is represented. In *Sox9^{fl/fl};Sox1^{Cre/+}* mutants, PROX1+ cells are accumulating in the lower blade (18.40 ± 2.29%) compared to controls (13.57 ± 1.29%, p=0.0187), and are reduced in the upper blade (8.57 ± 0.58%) compared to controls (13.13 ± 0.55%, Tukey's multiple comparison test p=0.0071, Two-way ANOVA interaction p=0.0387, row factor p<0.0001, column factor p=0.9991). A similar tendency was observed in *Sox9^{fl/fl};Nestin-Cre* mutants; however, it did not reached statistical significance. DG: dentate gyrus; DNE: dentate neuroepithelium; CH: cortical hem. Scale bar represent 50 μm in (Ai-ii) 100 μm in (Aiii-iv), (F) and (Ii-iii); 200 μm in (Iiv-vi).

The online version of this article includes the following source data and figure supplement(s) for figure 2:

**Source data 1.** Quantification of total PAX6, TBR2, and PROX1-expressing cells at E18.5 and P2 and their distribution along 1ry, 2ry, and 3ry matrices and/or within the forming dentate gyrus (DG).

**Figure supplement 1.** Initial emergence of intermediate progenitors (IPs) and differentiating granule neurons is not affected by *Sox9* deletion.

**Figure supplement 1—source data 1.** Quantification of total number of TBR2+ cells at E14.5 and PROX1+ cells at E16.5.

**Figure supplement 2.** *Sox9* deletion is not associated with increased cell death in the developing dentate gyrus (DG).

**Figure supplement 2—source data 1.** Quantification of Cleaved-Caspase+ cells in 1ry and 2ry matrix od P2 pups.

**Figure supplement 3.** *Sox9* deletion does not alter rate of neural progenitor proliferation, emergence, or differentiation toward a granule neuron fate.

**Figure supplement 3—source data 1.** Analysis of proliferation in PAX6, TBR2, and PROX1-expressing cells during dentate gyrus (DG) development.

data indicate that while total number, survival and emergence of granule neurons and their progenitors are not grossly affected in either *Sox9^{fl/fl};Sox1^{Cre/+}* or *Sox9^{fl/fl};Nestin-Cre* mutant embryos, postnatally there is a decrease in numbers of TBR2+ progenitors in *Sox9^{fl/fl};Sox1^{Cre/+}* animals.

While the total number of progenitors and granule neurons was unaffected in *Sox9* mutant embryos compared to controls, we observed an abnormal distribution of these cells along the three matrices (1ry, 2ry, and 3ry) (*Figure 2F,Iiv–vi,M*). At E18.5, we counted more TBR2+ cells in the 2ry matrix in *Sox9^{fl/fl};Nestin-Cre* mutants, apparently at the detriment of the 3ry matrix, where fewer cells were present in both *Nestin-Cre* and *Sox9^{fl/fl};Sox1^{Cre/+}* mutants compared to controls (*Figure 2Mi*). The misdistribution of TBR2+ cells become more evident post-natally, with fewer TBR2+ progenitors in the 3ry matrix of both mutant strains, but with more cells present in the 1ry matrix of *Sox9^{fl/fl};Sox1^{Cre/+}* embryos and in the 2ry matrix of *Sox9^{fl/fl};Nestin-Cre* mutants, compared to controls (*Figure 2Mii*). We analyzed in parallel the distribution of TBR2+EdU+ progenitors at P2 and confirmed this was abnormal in *Sox9^{fl/fl};Sox1^{Cre/+}* mutants, while not affected in *Sox9^{fl/fl};Nestin-Cre* mutants (*Figure 2—figure supplement 3G*). Similarly, in both *Sox9* mutants at P2, we observed a reduction in PROX1+ differentiating granule neurons in the 3ry matrix and, in parallel, significantly increased number of these cells in the 1ry matrix of *Sox9^{fl/fl};Sox1^{Cre/+}* mutants (*Figure 2Miii*). Conversely, 1ry-to-2ry matrix distributions of PAX6+ and PAX6+Ki67+ progenitors (*Figure 2LFigure 2—figure supplement 3C*) are not affected in either E14.5 or E16.5 *Sox9^{fl/fl};Sox1^{Cre/+}* mutants

compared to controls. This suggests that defective DG neuronal progenitor distribution in *Sox9* mutants arise between E16.5 and E18.5.

Furthermore, the distribution of PROX1+ cells within the 3ry matrix also appears disrupted at E18.5 in both mutants, albeit less severely in *Sox9*<sup>fl/fl</sup>;*Nestin-Cre* embryos (*Figure 2ii–iii*). To assess this defect, we quantified the number of PROX1+ cells in different bins ranging from the lower to the upper blade (*Figure 2N*). In controls, PROX1+ cells are equally distributed between the upper and lower blade. In contrast, in *Sox9*<sup>fl/fl</sup>;*Sox1*<sup>Cre/+</sup> mutants, these cells accumulate in the lower blade of the developing DG at the detriment of the upper blade. A similar tendency is observed in *Nestin-Cre* mutants; however, this does not reach statistical significance, suggesting a milder defect in this mutant strain. Furthermore, we noticed in the developing DG of both *Sox9* mutants, an ectopic cluster comprising a mix of both TBR2 and PROX1 expressing cells accumulating close to the ventricle since E18.5 (arrows in *Figure 2Fii,iii,v,vi,Iv,vi*).

Altogether these analyses show that starting from E18.5 progenitors and granule neurons are abnormally distributed in the developing DG of *Sox9* mutants, with *Sox1*<sup>Cre</sup> mutants being more severely affected, both along the 1ry-to-3ry matrix axis (*Figure 2M*) as well as within the forming DG (3ry matrix; *Figure 2N*). These observations suggest a defect in cell migration. The presence of an ectopic cluster of cells close to the ventricle is in agreement with this hypothesis, and we thus decided to characterize this further.

## Cell migration is impaired in the developing DG in absence of SOX9

We first analyzed the cellular composition of the ectopic cluster at P2 (*Figure 3A–E*). It is located next to the SOX2+ DNE (*Figure 3Aiv–vi*) and contains some SOX2+ progenitors and TBR2+ IPs (some of which are EdU+; *Figure 3A*). Moreover, we also observed ectopic expression of PROX1 (arrows in *Figure 3C*) indicating that some progenitors are locally differentiating in granule neurons at this stage. Their commitment toward a granular cell fate was already visible at E18.5, with cells in the ectopic cluster expressing NeuroD1 (*Roybon et al., 2009*) (arrows in *Figure 3D*). In conclusion, the ectopic cluster comprises cells at different stages of commitment toward the granule neuron fate.

In our initial analysis of TBR2+ progenitor distribution (*Figure 2M*), cells present in the ectopic cluster were included into the 2ry matrix numbers (ectopic cluster + migrating cells within the 2ry matrix). To quantify the size of the ectopic cluster, which can be indicative of the migration defect, we calculated the percentage of TBR2+ cells clustering in the ectopic cluster relative to the total number of TBR2+ cells in the 2ry matrix. The size of the ectopic cluster represents a significant proportion of TBR2+ cells within the 2ry matrix and was similar in both *Sox9*<sup>fl/fl</sup>;*Sox1*<sup>Cre/+</sup> (34.1 ± 2.3%) and *Sox9*<sup>fl/fl</sup>;*Nestin-Cre* (28.4 ± 3.1%) mutants at E18.5 (*Figure 3Ei*). While this proportion remained similar in *Sox9*<sup>fl/fl</sup>;*Sox1*<sup>Cre/+</sup> mutants at P2 (36.7 ± 5.3%), it was significantly decreased in *Sox9*<sup>fl/fl</sup>;*Nestin-Cre* animals at the same stage (17.9 ± 3.4%), in agreement with their milder phenotype (*Figure 3Ei*). A similar distribution was observed for newly generated progenitors at P2, with 40.13 ± 6.79% of TBR2+EdU+ cells of the 2ry matrix clustering in the ectopic cluster in *Sox9*<sup>fl/fl</sup>;*Sox1*<sup>Cre/+</sup> mutants, while this proportion is more than halved in *Sox9*<sup>fl/fl</sup>;*Nestin-Cre* counterparts (18.0 ± 3.5%; *Figure 3Eii*).

The presence of the ectopic cluster could be explained either by precocious differentiation of neuronal progenitors next to the ventricle or by their impaired migration toward the developing DG. To further analyze this aspect, lineage tracing of progenitors was performed using in utero electroporation in wild-type and *Sox9*<sup>fl/fl</sup>;*Sox1*<sup>Cre/+</sup> mutants. Progenitors facing the ventricle were electroporated at E15.5 with a plasmid-expressing dsRed and the distribution of dsRed+ cells within the three matrices was analyzed 7 days later at P2 (*Figure 3F,G*). The total number of dsRed+ cells within the developing DG at P2 was significantly lower in *Sox9*<sup>fl/fl</sup>;*Sox1*<sup>Cre/+</sup> mutants compared to controls (*Figure 3H*). We did not observe excess cell death in electroporated mutants compared either to the contralateral side, or to controls at this stage (*Figure 3—figure supplement 1*). However, electroporated mutant progenitors, whose survival may have been compromised by lack of SOX9, may have been lost earlier. In controls, 33.4 ± 7.26% of dsRed+ cells were PROX1+. In mutants, an average of 16.4 ± 15.11% of dsRed+ cells were PROX1+, and this was not significantly different from controls. However, this proportion was variable. This may be explained by variability in the domain targeted by the electroporation since our previous data (*Figure 2J*) showed that loss of SOX9 does not have an effect on granule neuron differentiation. We then analyzed the distribution of dsRed+

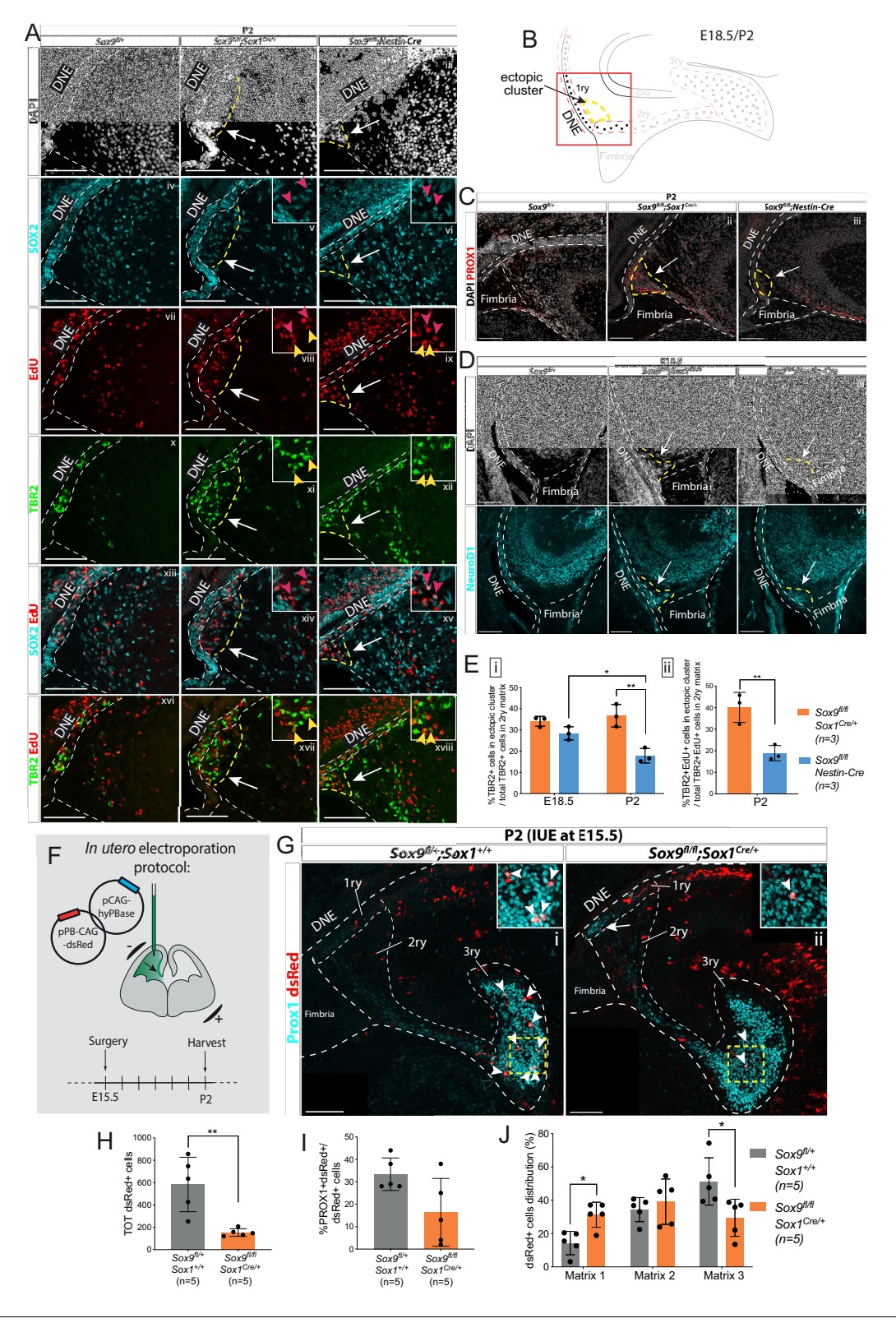

**Figure 3.** Ectopic accumulation of neuronal progenitors close to the ventricle suggests migratory defects in *Sox9* mutant dentate gyrus (DG). (**A**) Triple immunostaining for TBR2, SOX2, and EdU at P2 control, *Sox9*^fl/fl^;*Sox1*^Cre/+^ and *Sox9*^fl/fl^;*Nestin-Cre* brains. EdU was injected at E18.5. Insets show higher magnification of cells in ectopic cluster (schematized in **B**), magnified area is indicated by the white arrow. Yellow and pink arrowheads indicate TBR2+EdU+ and SOX2+EdU+ cells in the ectopic cluster, respectively. (**B**) Illustration showing location of ectopic cluster within the developing DG. (**C**–
*Figure 3 continued on next page*

Figure 3 continued

D) Immunofluorescences for the differentiation markers PROX1 (C) and NeuroD1 (D) in *Sox9^fl/fl^;Sox1^Cre/+^* and *Sox9^fl/fl^;Nestin-Cre* mutants compared to controls at P2 (C) and E18.5 (D), respectively. Both markers are expressed by cells within the ectopic cluster (arrows) in *Sox9* mutants. (E) Quantification of ectopic cluster size at E18.5 and P2 in *Sox9^fl/fl^;Sox1^Cre/+^* compared to *Sox9^fl/fl^;Nestin-Cre* mutants. The percentage of TBR2+ progenitors in ectopic cluster relative to total number of TBR2+ progenitors in 2ry matrix is represented. At E18 (i), the ectopic cluster size was comparable between *Sox9^fl/fl^; Sox1^Cre/+^* (34.11 ± 2.35%) and *Sox9^fl/fl^;Nestin-Cre* mutants (28.41 ± 3.10%). It then significantly decreases in *Sox9^fl/fl^;Nestin-Cre* mutants at P2 compared to E18.5 (17.87 ± 3.41%, *t* test p=0.0172) and *Sox9^fl/fl^;Sox1^Cre/+^* at the same stage (36.73 ± 5.30%, *t* test p=0.0061). In agreement with the smaller ectopic matrix size at P2, less newly formed TBR2+EdU+ progenitors (ii) were found in the ectopic cluster of *Sox9^fl/fl^;Nestin-Cre* mutants (18.90 ± 3.53%) compared to *Sox9^fl/fl^;Sox1^Cre/+^* (40.13 ± 6.97%, *t* test p=0.0084). (F) Schematic of in utero electroporation protocol. (G) Immunostaining for PROX1 and dsRed live fluorescence. Double-positive cells from the dashed yellow square are shown at higher magnification in the inset. (H) The total number of dsRed+ cells was significantly smaller in *Sox9^fl/fl^;Sox1^Cre/+^* mutants (153.60 ± 33.72) compared to controls (583.40 ± 243.76, p=0.0045 *t* test). (I) The proportion of dsRed+ on total PROX1+ cells was not significantly reduced in *Sox9* mutants (16.40 ± 15.11%) compared to controls (33.40 ± 7.27%). (J) Distribution of dsRed+ cells along the three matrices (as schematized in *Figure 2M*). We observed more dsRed+ cells in the 1ry matrix of *Sox9^fl/fl^; Sox1^Cre/+^* mutants compared to controls (31.33 ± 7.47% vs. 14.32±7.03%, *t* test p=0.0105) and less in the 3ry matrix (29.49 ± 11.13% vs. 51.27±14.20%, *t* test p=0.0287). DNE: dentate neuroepithelium; IUE: in utero electroporation. Scale bars represent 100 µm.

The online version of this article includes the following source data and figure supplement(s) for figure 3:

**Source data 1.** Quantification of ectopic matrix size at E18.5 and P2 and total number, differentiation, and distribution of dsRed+ cells at P2 upon in utero electroporation at E15.5.

**Figure supplement 1.** In utero electroporation does not compromise cell survival in the developing dentate gyrus of *Sox9* mutants.

cells in each matrix (represented by dotted white lines in *Figure 2G*). In P2 controls, the highest proportion of dsRed+ cells was observed in the 3ry matrix (51.27 ± 14.20%), demonstrating that an important fraction of E15.5 electroporated progenitors had given rise to migrating granule neurons that successfully reached their destination in the developing DG (*Figure 3G,J*, arrowheads in G). In contrast, in *Sox9^fl/fl^;Sox1^Cre/+^* mutants, the highest proportion of dsRed+ cells was found in the 2ry matrix (39.18 ± 13.59%) and the fraction of cells remaining in the 1ry matrix was significantly higher than in controls (*Figure 3G,J*). This suggests that, in *Sox9* mutants, a proportion of electroporated progenitors remained trapped near the DNE (arrow in *Figure 3Gii*). These results are thus in agreement with impaired neuronal progenitor migration in the developing DG of *Sox9* mutants. We then investigated the origin of this phenotype by examining known molecular mechanisms regulating this process.

## Delayed induction of GFAP+ glial scaffold and its progenitors in absence of SOX9

Expression of chemokines (*Reln*, *Cxcl12*) and their receptors (*Vldlr*, *Cxcr4*) known to be involved in early stages of DG progenitor migration (*Frotscher et al., 2003*; *Mimura-Yamamoto et al., 2017*), is not significantly different in *Sox9* mutant E12.5 dissected archicortices compared to controls (*Figure 4—figure supplement 1A*). These results are consistent with the absence of early migration defects in *Sox9* mutants (*Figure 2L*). Similarly, at E18.5, REELIN expression pattern and intensity appeared unchanged in both *Sox9* mutants compared to controls (*Figure 4—figure supplement 1B*) further suggesting that Cajal-Retzius (CR) cells are not affected by loss of *Sox9*.

In addition to CR cells, a GFAP-expressing glial scaffold has been previously suggested to support DG progenitor migration during embryonic development (*Barry et al., 2008*). We thus examined expression of GFAP and observed a strongly positive scaffold in control samples from E18.5 connecting the DNE to the forming DG, through the fimbria (*Figure 4Ai,B*). In contrast, this is almost absent in *Sox9^fl/fl^;Sox1^Cre/+^* (*Figure 4Aii*) but only partially affected in *Sox9^fl/fl^;Nestin-Cre* embryos (quantified in *Figure 4C*), where the supragranular glial scaffold is missing, but the fimbrial glial scaffold is still visible (inset *Figure 4Aiii*; see schematic B). GFAP expression and scaffold structure in both *Sox9* mutants partially recover by P2 (*Figure 4Aiv–vi*, quantified in C), suggesting that absence of SOX9 might only delay scaffold formation, and that either compensatory or independent mechanisms may allow recovery early post-natally. However, the impact on DG morphogenesis is permanent.

The supporting role of the glial scaffold for DG neuronal progenitor migration has never been formally demonstrated. We aimed to further assess its functionality in this context by closely examining the distribution of TBR2 progenitors in relation to the GFAP+ scaffold at E18.5 when the scaffold

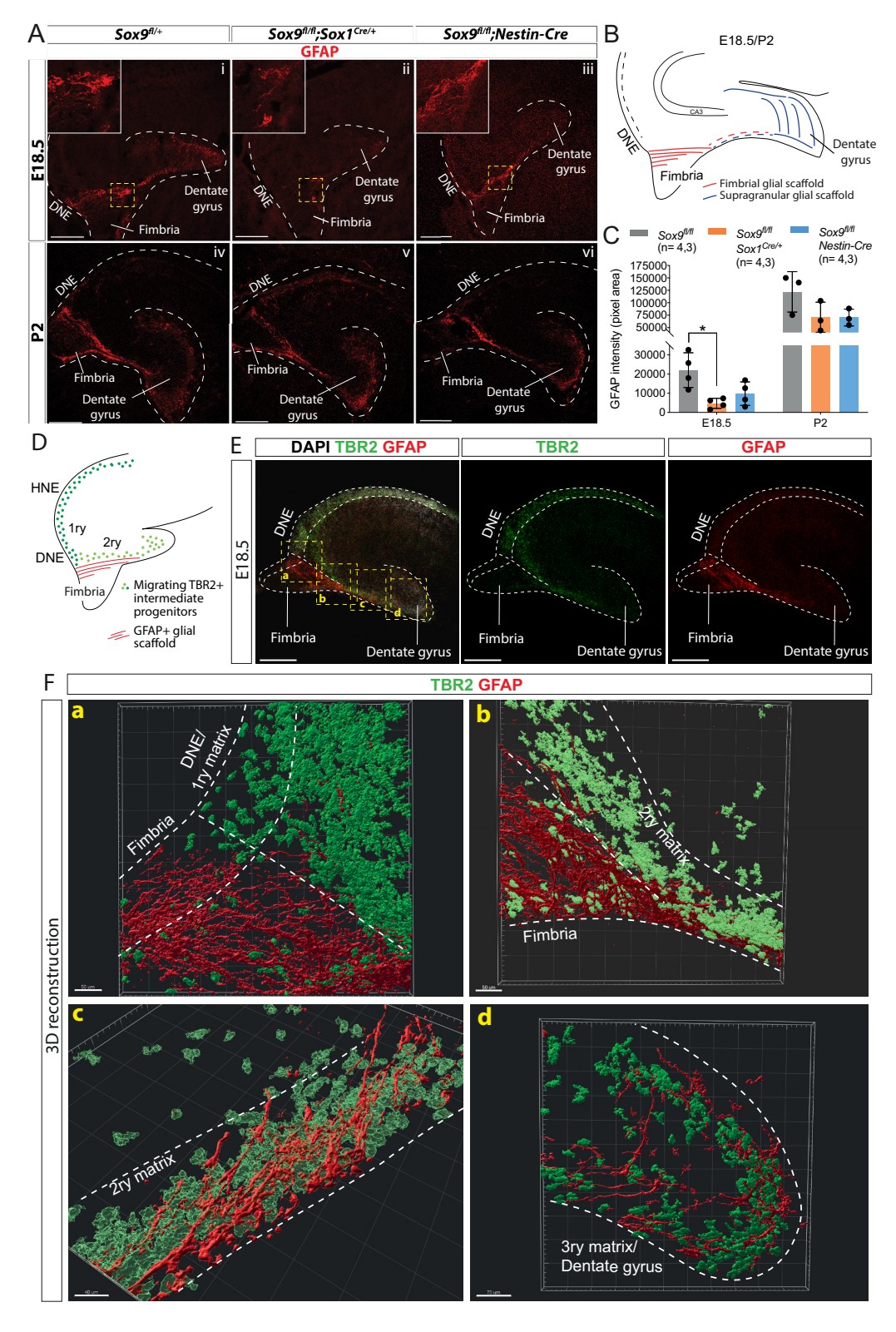

**Figure 4.** Delay in formation of the glial scaffold in *Sox9* mutants may explain progenitor migration defects. (**A–C**) Analysis of glial scaffold formation. (**A**) Immunofluorescence for GFAP on E18.5 (Ai-iii) and P2 (Aiv-vi) control, *Sox9^{fl/fl};Sox1^{Cre/+}* and *Sox9^{fl/fl};Nestin-Cre* brains showing GFAP reduction in both mutants at E18.5. Dashed line delineates the developing dentate gyrus (DG) area, yellow dashed squares indicate areas magnified in insets. (**B**) Representation of the glial scaffold (red lines) in DG. (**C**) GFAP immunofluorescence quantification (pixel area). At E18.5, GFAP expression was

*Figure 4 continued on next page*

*Figure 4 continued*

significantly reduced in *Sox9*<sup></sup>*fl/fl;Sox1*<sup>*Cre*</sup> mutants (4745.17 ± 2609.79) compared to controls (22069.97 ± 9082.47, p=0.01120), while not in *Sox9*<sup>*fl/fl*</sup>;*Nestin-Cre* mutants (9803.93 ± 6141.10, Tukey's multiple comparison test p=0.06090, ANOVA p=0.0121). At P2, GFAP expression is recovered in both *Sox9* mutants compared to controls. (D–F) 3D reconstruction of control E18.5 embryos double immunostained for TBR2 and GFAP, (E) Representative control 10x single-plane confocal images of sections processed for 3D reconstruction (schematized in D; yellow dashed squares indicate processed regions shown in F). (F) Snapshots from 3D reconstruction show that the fimbrial scaffold and 1ry matrix progenitors are initially separated (a). 2ry matrix migrating progenitors then start to intermingle with GFAP+ fibers as the scaffold elongates from the fimbria (b,c). 3ry matrix progenitors are also distributed within the supragranular scaffold within the developing DG (d). Movies of all 3D reconstructions are available in the supplementary material (*Videos 1–4*). DNE: dentate neuroepithelium. Scale bars represent 200 µm.

The online version of this article includes the following source data and figure supplement(s) for figure 4:

**Source data 1.** Quantification of GFAP expression at E18.5 and P2.

**Figure supplement 1.** Migratory clues secreted by Cajal-Retzius cells and required during dentate gyrus (DG) development are not affected in *Sox9* mutants.

**Figure supplement 1—source data 1.** Quantification of *Cxcr4, Vldr, Cxcl12,* and *Reeln* expression with qPCR in E12.5 dissected ARK.

first appears (*Figure 4D–F*, *Videos 1–4*). Close to the DNE, the GFAP+ fimbrial scaffold appears well separated from both the intermediate DNE progenitors and those migrating in the 2ry matrix (*Figure 4Fa*). Along the migratory stream in the 2ry matrix, progenitors start being more closely associated with the GFAP+ scaffold (*Figure 4Fb*). Finally, from the distal part of the fimbria, progenitors and scaffold appear completely intermingled (*Figure 4Fc*). We observe a similar association in the 3ry matrix between IPs and the GFAP supragranular scaffold (*Figure 4Fd*). The position of IPs relative to the glial scaffold suggest close contacts between these cell populations, supporting a functional role for the glial scaffold in promoting progenitor migration, particularly within the 2ry matrix. Consequently, delayed formation of the glial scaffold in *Sox9* mutants is likely implicated in the defective migration of DG progenitor.

Because SOX9 directly regulates the expression of *Gfap* in the developing spinal cord (*Kang et al., 2012*), absence of GFAP expression in *Sox9* mutants may simply reflect downregulation of the gene. Therefore, to confirm the transient defect in glial scaffold formation, we examined the expression of ALDH1L1, an astrocyte-specific marker (*Cahoy et al., 2008*). At E18.5, ALDH1L1 expression pattern overlaps with that of GFAP, particularly within the fimbria where we see many GFAP+; ALDH1L1+ fibers (arrowheads in inset b in *Figure 5Ai*). In contrast, we observe GFAP +ALDH1L1- fibers around the developing DG (arrowheads in inset a in *Figure 5Ai*). At E16.5, before upregulation of GFAP, the ALDH1L1 expression pattern is reminiscent of that at E18.5, as mostly confined to the fimbria (*Figure 5Aii*). ALDH1L1+ cells are found as early as E13.5 in the archicortex (*Figure 5Aiii*) and also at this stage, they are specifically localized in the LEF1-negative;SOX2<sup>high</sup> CH

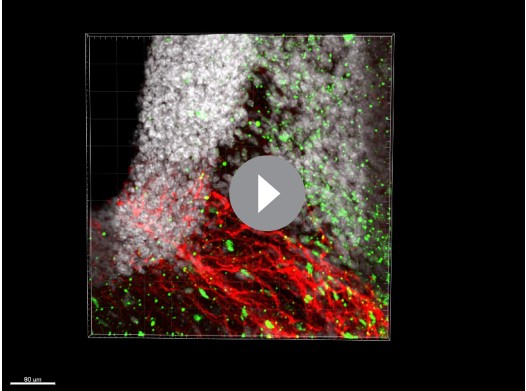

**Video 1.** Movie of 3D reconstruction of progenitors at the primary matrix level. GFAP+ fimbrial scaffold (red) and 1ry matrix TBR2+ progenitors (in green) are initially separated.

https://elifesciences.org/articles/63904#video1

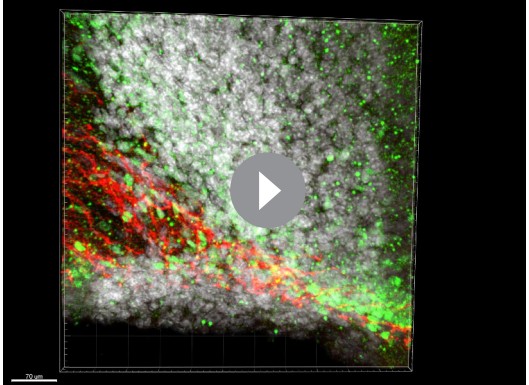

**Video 2.** Movie of 3D reconstruction of migrating progenitors at the secondary matrix level. TBR2+ migrating progenitors (green) in the 2ry matrix start to intermingle with GFAP+ fibers (red).

https://elifesciences.org/articles/63904#video2

(*Sugiyama et al., 2013*; *Figure 5B,Ci–ii*). The astrocytic nature of ALDH1L1+ cells was further confirmed by stainings for BLBP and GLAST, known markers of astrocytic progenitors (*Nagao et al., 2016*) which are also present within the LEF1-negative; SOX2$^{high}$ CH (*Figure 5Ciii,iv*). Altogether, these results indicate that astrocytic progenitors are confined to the CH/fimbria throughout development suggesting they might later give rise to the fimbrial glial scaffold.

We then analyzed whether ALDH1L1+ cells were affected by absence of SOX9. Strikingly, there was a dramatic reduction in their number in *Sox9$^{fl/fl}$;Sox1$^{Cre/+}$* mutants compared to controls, both at E13.5 and E16.5 (*Figure 5D,E*). Accordingly, *Aldh1l1* expression was significantly reduced in dissected archicortices of *Sox9$^{fl/fl}$;Sox1$^{Cre/+}$* E12.5 embryos compared to controls (*Figure 5F*). This is consistent with a requirement for SOX9 for the emergence of astrocytic ALDH1L1+ progenitors, and consequently formation of the GFAP+ glial scaffold.

In contrast to *Sox9$^{fl/fl}$;Sox1$^{Cre/+}$* mutants, ALDH1L1+ cell number was unaffected in *Sox9$^{fl/fl}$;Nestin-Cre* (*Figure 5D–F*). Because *Nestin-Cre*-mediated recombination occurs within the CH in a salt and pepper manner, SOX9 and ALDH1L1 expression patterns were analyzed in this region in both *Sox9* mutants. At E13.5, ALDH1L1+ cells were expressing SOX9 in the CH of *Sox9$^{fl/fl}$;Nestin-Cre* embryos (*Figure 5Gii,vi*), and this was still observed at E16.5 (*Figure 5Giv,viii*). Interestingly, some rare SOX9+ cells were also present in the CH of E13.5 *Sox9$^{fl/fl}$;Sox1$^{Cre/+}$* mutants, and some were ALDH1L1+ (*Figure 5Gi,v*). These results suggest that in the CH of *Sox9* mutants, ALDH1L1+ cells may only arise from SOX9+ progenitors that escaped Cre recombination, which are present in higher numbers in *Sox9$^{fl/fl}$;Nestin-Cre* compared to *Sox9$^{fl/fl}$;Sox1$^{Cre/+}$* mutants (schematized in *Figure 5H*). Moreover, the correlation between the extent of ALDH1L1+ cells and fimbrial glial scaffold loss with the severity of the progenitor migration defect in *Sox9$^{fl/fl}$;Sox1$^{Cre/+}$* versus *Sox9$^{fl/fl}$;Nestin-Cre* mutants, further suggests a supporting migratory role of the scaffold.

Finally, in the developing spinal cord, the expression of the transcription factors NF1A/B are regulated by SOX9 and this is important for astrocytic differentiation (*Kang et al., 2012*). We thus examined expression of NF1A and B in *Sox9$^{fl/fl}$;Sox1$^{Cre/+}$* E12.5 embryos. Both genes are expressed in the archicortex in control embryos. Loss of SOX9 does not affect either NF1A/B protein or transcript levels (*Figure 5—figure supplement 1*). We conclude that distinct molecular mechanisms downstream of SOX9 must underlie astrocytic specification in different domains of the CNS.

## CH-specific deletion of *Sox9* using *Wnt3a$^{iresCre/+}$* impairs fimbrial glial scaffold formation and compromises granule neuron progenitor migration

Because both *Sox1$^{Cre/+}$* and *Nestin-Cre* are also active in the DNE (*Figure 1G*), cell autonomous defects could contribute to defective granule neuron progenitor migration. To examine this possibility and to confirm the requirement for SOX9 in the CH for formation of the fimbrial glial scaffold, CH-specific deletion of *Sox9* was performed using *Wnt3a$^{iresCre}$* (*Yoshida et al., 2006*). First, we confirmed *Wnt3a$^{iresCre}$* specificity to the CH by performing lineage tracing. In *Wnt3a$^{iresCre}$;R26R$^{eYFP/+}$* embryos, eYFP staining is mostly confined to the LEF1- CH at E12.5 (*Figure 6Ai–iii*) and to CH-derived REELIN+ CR cells both around the DG (Figure S.12.Ai-iv) and in the outer layer of the cortex (*Figure 6—figure supplement 1Av–viii*). Because a few YFP+ cells were observed in the LEF1+ DNE, suggesting partial *Wnt3a$^{iresCre}$* recombination in the DNE (arrowheads in *Figure 6Ai–iii*), we analyzed YFP expression in granule neurons at P2. Only 6.45 ± 1% of PROX1+ granule neurons were YFP+ at this stage (arrowheads in *Figure 6Bi–iii*), suggesting *Wnt3a$^{iresCre}$* is a suitable Cre driver for CH-specific deletion of *Sox9*.

ALDH1L1+ and BLBP+ astrocytic progenitors in the CH were also recombined by *Wnt3a$^{iresCre}$* (*Figure 6Aiv–ix*). At P2, the fimbrial glial scaffold is entirely eYFP+ (arrowheads in *Figure 6Biv–vi*). We also observed some GFAP+;eYFP- filaments in the DNE that may represent DNE-derived RGCs (arrows in *Figure 6Biv–vi*). Conversely, the supragranular glial scaffold is made of both eYFP+ and eYFP- fibers suggesting a dual DNE and CH origin (arrowheads and arrows in *Figure 6Bvii–ix*, respectively). Altogether these results suggest that the fimbrial glial scaffold is entirely derived from CH ALDH1L1+ astrocytic progenitors, while the supragranular glial scaffold only partially originates from CH. Additionally, we did not observe any PDGFRa+ oligodendrocyte precursor cells (OPCs) in the progeny of CH *Wnt3a$^{iresCre}$* cells (*Figure 6—figure supplement 1B*) arguing in favor of an astrocytic, rather than radial-glial, nature of the scaffold.

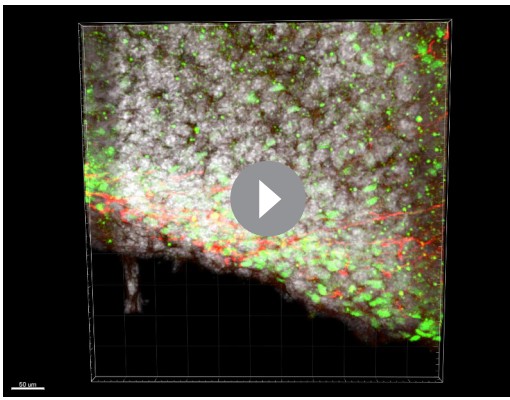

**Video 3.** Movie of 3D reconstruction of migrating progenitors as the secondary matrix elongates. Intermingling of TBR+ migrating progenitors (green) and GFAP+ fibers (red) in the distal part of the 2ry matrix.

https://elifesciences.org/articles/63904#video3

*Sox9^{fl/fl}*;*Wnt3a^{iresCre/+}* mutants were then generated. While we were able to harvest mutant embryos until E18.5, animals died shortly after birth, precluding any postnatal analyses. *Wnt3a* is widely expressed in embryonic mesoderm precursors (*Takada et al., 1994*) and deletion of *Sox9* in the embryonic heart, skeleton, pancreas, and kidney is known to result in postnatal lethality (*Seymour et al., 2007*; *Akiyama et al., 2004*; *Reginensi et al., 2011*). In *Sox9^{fl/fl}*;*Wnt3a^{iresCre/+}* mutant CNS, SOX9 is absent specifically in the CH at E13.5 (*Figure 6Ci–iv*). Importantly, we also observe a 50% reduction of ALDH1L1+ cells in this area in E13.5 mutants compared to controls (*Figure 6Cv,vi*; quantified in D). Interestingly, at E18.5, the GFAP+ fimbrial glial scaffold is exclusively compromised in *Sox9^{fl/fl}*;*Wnt3a^{iresCre/+}* mutants (star in *Figure 6Cviii*), while the supragranular one is unaffected. Quantification was performed by measuring GFAP immunofluorescence separately in the fimbrial and supragranular scaffold (schematic in *Figure 6E*). This analysis clearly shows that GFAP expression is significantly reduced in the fimbrial scaffold, in *Sox9^{fl/fl}*;*Sox1-Cre/+* and *Sox9^{fl/fl}*;*Wnt3a^{iresCre/+}* mutants, compared to controls, but not when *Nestin-Cre* is used to delete *Sox9*. (*Figure 6F*). Conversely, GFAP expression in the supragranular scaffold, is reduced in *Sox9^{fl/fl}*;*Sox1^{Cre/+}* and *Sox9^{fl/fl}*;*Nestin-Cre* mutants compared to controls, but not in *Sox9^{fl/fl}*;*Wnt3a^{iresCre/+}* mutants. These results are consistent with the differential activity pattern of the Cre drivers. Moreover, they confirm a role for SOX9 in the CH for specification of the ALDH1L1+ astrocytic progenitors giving rise to GFAP+ fimbrial glial scaffold. The normal appearance of the supragranular glial scaffold in *Sox9^{fl/fl}*;*Wnt3a^{iresCre/+}* mutants is consistent with the observation that it may have a dual CH and DNE origin (*Figure 6Bvii–ix*).

We then analyzed granule neurons and their progenitors in *Sox9^{fl/fl}*;*Wnt3a^{iresCre/+}* mutants. While DG morphology at E18.5 is not affected in these mutants, an ectopic cluster is clearly visible next to the DNE (*Figure 7—figure supplement 1*). At E18.5, the total number of PROX1+ granule neurons and TBR2+ progenitors is unchanged in mutants compared to controls (*Figure 7A,B* and *Figure 7E, F* respectively), similarly to what we observed in *Sox9^{fl/fl}*;*Sox1^{Cre/+}* mutants. Therefore, CH-specific deletion of *Sox9* does not affect progenitor formation and differentiation. We then analyzed the distribution of granule neurons. At E18.5, PROX1+ cell distribution in the 3ry matrix is unaffected in *Sox9^{fl/fl}*;*Wnt3a^{iresCre/+}* mutants compared to controls (*Figure 7A,C*). This is consistent with the supragranular glial scaffold not being affected upon CH-specific deletion of *Sox9* (*Figure 6Fviii*). In fact, repartition of granule neurons in the upper blade and lower blade at E18.5 is exclusively compromised in *Sox9^{fl/fl}*; *Sox1^{Cre/+}* and *Sox9^{fl/fl}*;*Nestin-Cre* mutants (*Figure 7D*). Because in both *Nestin-* and *Sox1-Cre* models SOX9 is absent in the DNE, these results, together with previous observations (*Li et al., 2009*; *Heng et al., 2012*), are in accord with a DNE contribution for the formation of the supragranular glial scaffold.

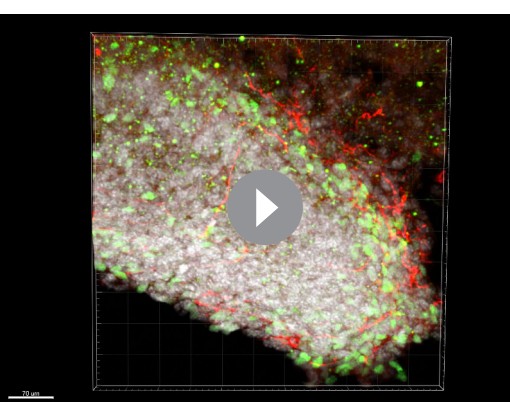

**Video 4.** Movie of 3D reconstruction of migrating progenitors at the tertiary matrix level. Distribution of TBR2+ progenitors (green) within the GFAP+ supragranular scaffold (red).

https://elifesciences.org/articles/63904#video4

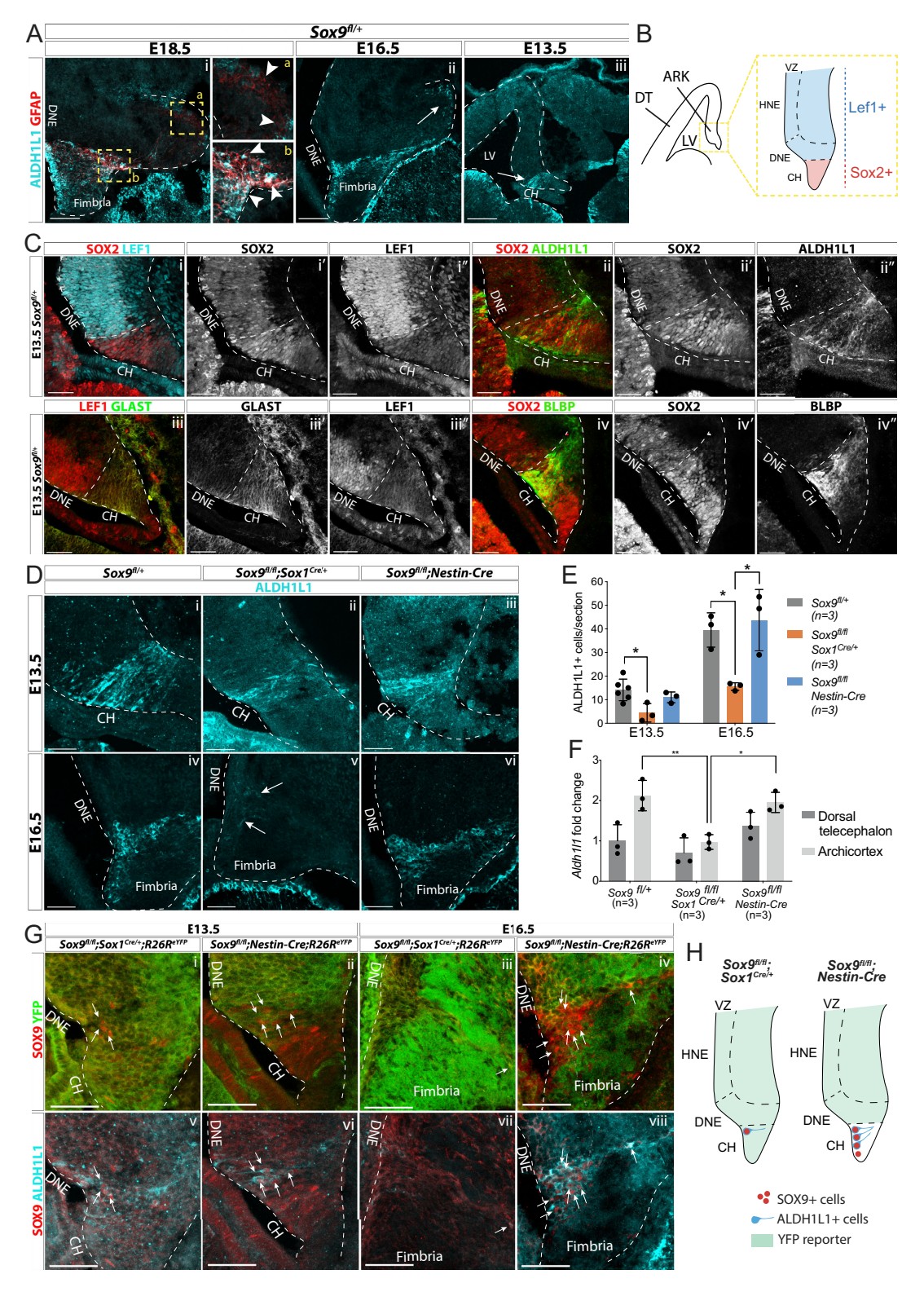

**Figure 5.** Emergence of astrocytic progenitor in the CH is affected in *Sox9* mutants according to levels of Cre activity in this region. (**A**) Double immunostaining for ALDH1L1 and GFAP in *Sox9*<sup>fl/+</sup> embryos at E18.5 (i), E16.5 (ii), and E13.5 (iii). ALDH1L1 and GFAP are co-expressed at E18.5 (i) in the fimbria (**b** insets on the right) but not around the forming dentate gyrus (DG; **a** insets on the right). Earlier, at E16.5 (ii), ALDH1L1, but not GFAP, is expressed in a similar pattern, in the fimbria and in a few cells around the forming DG (arrow in Aii), and as early as E13.5 in the archicortex (arrow in

*Figure 5 continued on next page*

Figure 5 continued

Aiii). (B, C) Double immunostainings for SOX2;LEF1 (i), SOX2;ALDH1L1 (ii) at E13.5. SOX2 and LEF1 mutually exclusive expression patterns delineate the LEF1-SOX2$^{high}$ CH and LEF1+SOX2$^{low}$ DNE (schematized in B). ALDH1L1+ cells are exclusively located in the SOX2$^{high}$ CH. Double immunostaining for GLAST;LEF1 (iii) and BLBP;SOX2 (iv) show a similar pattern of expression of the two astrocytic markers GLAST and BLBP in the LEF1-;SOX2$^{low}$ CH, further suggesting ALDH1L1+ cells astrocytic nature. (D–E) Immunostaining (D) and quantification (F) of ALDH1L1+ cells in *Sox9* mutants at E13.5 (i-iii) and E16.5 (iv-vi) compared to controls. White arrows in Fv indicate rare ALDH1L1+ cells found in *Sox9$^{fl/fl}$;Sox1$^{Cre}$* mutants. The number of ALDH1L1+ cells was significantly reduced in *Sox9$^{fl/fl}$;Sox1$^{Cre}$* mutants compared to controls, both at E13.5 (4.43 ± 3.93 vs. 14.14 ± 4.58, p=0.0193) and E16.5 (15.58 ± 1.62 vs. 39.54 ± 7.27, p=0.0338), while it was unaffected in *Sox9$^{fl/fl}$;Nestin-Cre* mutants (E13.5: 11.00 ± 2.29, p=0.5373, E16.5: 43.73 ± 13.00, p=0.8288, Tukey's multiple comparison test, ANOVA p=0.0242 and p=0.0147). (F) Analysis of *Aldh1l1* expression levels by qPCR from dissected DT and ARK of E12.5 *Sox9$^{fl/fl}$;Sox1$^{Cre}$*, *Sox9$^{fl/fl}$;Nestin-Cre* and control embryos. *Aldh1l1* expression was significantly reduced in the ARK of *Sox9$^{fl/fl}$;Sox1$^{Cre}$* compared to both controls (p=0.0028) and *Sox9$^{fl/fl}$;Nestin-Cre* mutants (p=0.0047, Tukey's multiple comparison test, ANOVA p=0.0021). (G,H) Triple immunostaining for YFP, SOX9 and ALDH1L1 at E13.5 (i,ii,v,vi) and E16.5 (ii,iv,vii,viii) in *Sox9$^{fl/fl}$;Sox1$^{Cre}$* and *Sox9$^{fl/fl}$;Nestin-Cre* mutants. A few double-positive SOX9;ALDH1L1 cells are detected in the CH of both *Sox9* mutants (white arrows, schematized in H). More of these are present in *Sox9$^{fl/fl}$;Nestin-Cre* compared to *Sox9$^{fl/fl}$;Sox1$^{Cre/+}$* mutants due differential Cre activity as shown by the *R26R$^{eYFP}$* reporter expression. LV: lateral ventricle; DNE: dentate neuroepithelium; CH: cortical hem; DT: dorsal telencephalon; ARK: archicortex; HNE: hippocampal neuroepithelium; VZ: ventricular zone. Scale bars represent 200 μm in (Aiii), 100 μm in (Ai,ii), and 50 μm in (C), (D), and (G).

The online version of this article includes the following source data and figure supplement(s) for figure 5:

**Source data 1.** Quantification of ALDH1L1+ cells at E13.5 and E16.5 and of *Aldh1l1* expression with qPCR in E12.5 DT and ARK separately.
**Figure supplement 1.** Early CNS-specific deletion of *Sox9* does not affect NF1A/B expression in the forebrain.
**Figure supplement 1—source data 1.** Quantification of *Nfia* and *Nfib* expression with qPCR in E12.5 DT and ARK separately.

We then examined distribution of TBR2+ progenitors in the three matrices at E18.5. We observe that more cells accumulate in the 2ry matrix of *Sox9$^{fl/fl}$;Wnt3a$^{iresCre/+}$* mutants compared to controls (*Figure 7G*). This abnormal distribution is reminiscent of that seen in *Sox9$^{fl/fl}$;Sox1$^{Cre/+}$* mutants (*Figure 2F,Mi*). Furthermore, progenitors in *Sox9$^{fl/fl}$;Wnt3a$^{iresCre/+}$* mutants form an ectopic cluster close to the ventricle (yellow dashed line in *Figure 7Eii*), with a size comparable to that seen in both *Sox9$^{fl/fl}$;Sox1$^{Cre/+}$* and *Sox9$^{fl/fl}$;Nestin-Cre* mutants at the same stage (*Figure 7H*). The ectopic cluster comprises differentiating neurons, with some cells expressing NeuroD1 (arrow in *Figure 7J*) as previously observed in *Sox9$^{fl/fl}$;Sox1$^{Cre/+}$* and *Sox9$^{fl/fl}$;Nestin-Cre* mutants (*Figure 3D*). Defective localization is not due to cell autonomous defects, because cells accumulating in the ectopic cluster are reporter negative in E18.5 *Sox9$^{fl/fl}$;Wnt3a$^{iresCre/+}$;R26R$^{eYFP}$* mutants (*Figure 7Kv–vii*), indicating their precursors were not deleted for *Sox9*. Altogether, these results strongly suggest that SOX9 is required in the CH for astrocytic specification and subsequent fimbrial glial scaffold formation. Furthermore, they demonstrate that defective localization of neuronal progenitors is a non-cell-autonomous defect, consistent with a lack of migratory support by the defective CH-derived fimbrial glial scaffold (*Figure 7Kviii*).

## Discussion

Using conditional deletion approaches, we have dissected the role of SOX9 during DG development. Differential patterns of gene deletion in the archicortex had marked consequences on adult DG morphology, and functionality, and we were consequently able to establish that SOX9 is required for proper DG morphogenesis. More precisely, our results highlight the crucial role of SOX9 for timely induction of the gliogenic switch in the CH, allowing emergence of astrocytic progenitors and subsequent formation of the fimbrial glial scaffold. Moreover, we show that this portion of the scaffold, which is closely associated with granule neuron progenitors, is necessary for their migration toward the forming DG. Furthermore, while partial recovery of the scaffold is observed early post-natally in *Sox9* mutants, DG morphogenesis is permanently affected, strongly suggesting that the glial scaffold is required, albeit transiently. In conclusion, our study unravels the cascade of events orchestrating the establishment of supportive astrocytic-neural interactions required for DG morphogenesis, highlighting its sensitivity to timing and dependence on SOX9.

### Dual origin, function and nature of the DG glial scaffold

Lineage tracing experiments using *Wnt3a$^{iresCre}$* demonstrate here that the fimbrial part of the glial scaffold has a CH origin (*Figure 8*). Our analyses of the fimbrial scaffold and of the defects observed

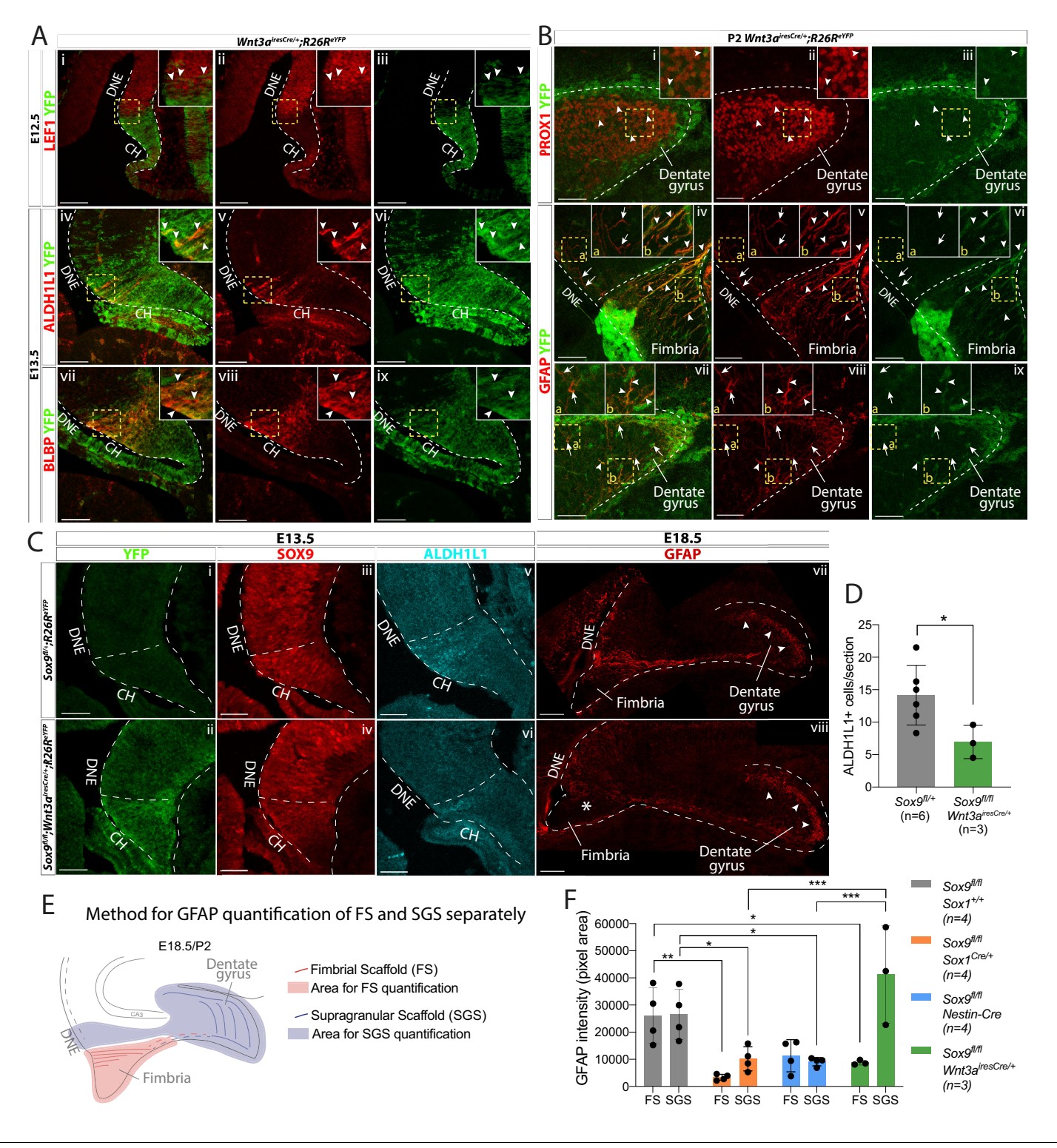

**Figure 6.** CH-specific deletion of *Sox9* using *Wnt3a^iresCre* compromises glial scaffold formation exclusively within the fimbria. (**A,B**) Analysis of *Wnt3a^iresCre* recombination pattern in the archicortex. (**A**) Double immunofluorescence for YFP with LEF1 (i-iii), ALDH1L1 (iv-vi) and BLBP (vii-ix) in *Wnt3a^iresCre/+;R26R^eYFP* embryos at E12.5 (i-iii) and E13.5 (iv-ix). Insets are magnified areas from yellow dashed boxes. Cre recombination is mostly observed in the LEF1- CH, however a few YFP+ cells are seen in the LEF1+ DNE (i-iii; arrowheads in magnified inset). ALDH1L1+;BLBP+ astrocytic progenitors express YFP in *Wnt3a^iresCre//+;R26R^eYFP* embryos (iv-xi; arrowheads in magnified inset), also confirming their CH origin. (**B**) Double immunofluorescences for PROX1;YFP (Bi-iii) and GFAP;YFP (iv,ix) in P2 *Wnt3a^iresCre/+;R26R^eYFP* embryos. Insets are magnified areas from yellow dashed

*Figure 6 continued on next page*

*Figure 6 continued*

boxes. Cells that have undergone Cre recombination are mostly GFAP+ and PROX1-, in agreement with a CHspecific recombination pattern (**A**). Arrowheads in Bi-iii and insets indicate some rare YFP+PROX1+ cells in the DG representing 6.45 ± 1.00% of PROX1+ cells. Arrows and arrowheads in Biv,ix and insets indicates respectively YFP-GFAP+ and YFP+GFAP+ fibers in the DNE/fimbria (Biv-vi) and around the DG (Bvii-ix), indicating the GFAP+ glial scaffold around the DG only partially originates from the CH. (**C,D**) Immunostainings and quantification for YFP (Ci,ii); SOX9 (Ciii,iv); ALDH1L1 (Cv, vi) at E13.5 and GFAP at E18.5 in $Sox9^{fl/fl};Wnt3a^{iresCre/+};R26R^{eYFP}$ mutant compared to controls. In E13.5 archicortices of $Sox9^{fl/fl};Wnt3a^{iresCre/+}$ mutant and control embryos the CH specific deletion of SOX9 is confirmed. The number of ALDH1L1+ cells is significantly reduced (**D**) in $Sox9^{fl/fl};Wnt3a^{iresCre/+}$ mutants (6.95 ± 2.56) compared to controls (14.14 ± 4.58, $t$-test p=0.022). At E18.5, the GFAP+ glial scaffold is affected exclusively within the fimbria (star in Cviii) and not around the DG (arrowheads in Cvii-viii). (**E, F**) Quantification of GFAP immunofluorescence as pixel area in the FS and SGS separately, based on morphology from DAPI as shown in (**F**). GFAP expression is significantly lower in the FS of both $Sox9^{fl/fl};Sox1^{Cre}$ (3357.78 ± 1101.38, p=0.0029) and $Sox9^{fl/fl};Wnt3a^{iresCre/+}$ (8783.77 ± 898.29, p=0.043) mutants compared to controls (26114.39 ± 10208.45) but not in $Sox9^{fl/fl};Nestin-Cre$ mutants (11304.45 ± 5919.25, Sidak multiple comparison test, Two-way ANOVA interaction p=0.0027). Conversely, GFAP expression in the SGS is significantly lower in $Sox9^{fl/fl};Sox1^{Cre}$ (10166.42 ± 4443.82, p=0.0377) and $Sox9^{fl/fl};Nestin-Cre$ mutants (9096.35 ± 1545.00, p=0.0249) compared to controls (26560.38 ± 9242.99) but not in $Sox9^{fl/fl};Wnt3a^{iresCre/+}$ mutants (41270.70 ± 18028.47; Sidak multiple comparison test, Two-way ANOVA interaction p=0.0027). CH: cortical hem; DNE: dentate neuroepithelium; DG: dentate gyrus; FS: fimbrial scaffold; SGS: supragranular scaffold. Scale bars represent 100 µm in (Ai-iii) and (Cvii-viii); 50 µm in (Aiv-ix), (**B**), (Ci-vi).

The online version of this article includes the following source data and figure supplement(s) for figure 6:

**Source data 1.** Analysis of ALDH1L1+ cells at E13.5 and GFAP expression at E18.5 in $Sox9^{fl/fl};Wnt3a^{iresCre/+}$ mutants compared to controls.
**Figure supplement 1.** Lineage-tracing analysis of CH-derived cells in $Wnt3a^{iresCre/+};R26R^{eYFP}$ pups.

in its absence, show that it supports neuronal progenitor migration from 1ry-to-3ry matrix (*Figure 8F*). This had been suggested previously, but without direct evidence (*Li et al., 2009*; *Barry et al., 2008*). In addition, because in *Sox9* mutants, early PAX6+ progenitor migration is not affected and the ectopic cluster appears next to the DNE from E18.5, the fimbrial glial scaffold only become indispensable for migration at late stages of DG development. We hypothesize this may be explained by the increasing distance between DNE and the forming DG, as development proceeds. Loss of $Sox9^{fl/fl};Wnt3a^{iresCre/+}$ mutants after E18.5, impedes any analysis on long-term effect of fimbrial scaffold disruption on DG development.

In agreement with a DNE origin for the distal part of the scaffold (*Li et al., 2009*; *Heng et al., 2012*) (*Figure 8*), the supragranular glial scaffold, was mostly unlabeled in our $Wnt3a^{iresCre}$ lineage-tracing experiments. However, the presence of some labeled fibers suggests a CH contribution. Beside a different cellular composition, the supragranular scaffold has also been suggested to have a distinct function from the fimbrial one, where the former guides granule neuron migration within DG upper and lower blades (*Heng et al., 2012*) (*Figure 8F*). In agreement, in $Sox9^{fl/fl};Wnt3a^{iresCre/+}$ mutants, where SOX9 is deleted exclusively in the CH (*Figure 8D*), the supragranular scaffold and consequently distribution of granule neurons in the 3ry matrix of the developing DG, appear normal. In contrast, we observe a transient impairment of the supragranular scaffold in both $Sox9^{fl/fl};Sox1-Cre/+$ (*Figure 8B*) and, to a lesser extent, in $Sox9^{fl/fl};Nestin-Cre$ mutants (*Figure 8C*), along with an altered upper/lower blade distribution of granule neurons. These results are in accord with a different role for this part of the scaffold as well as its predominant DNE origin since both *Sox1* and *Nestin-Cre* drivers are active in this domain. However, we observe a milder alteration in DG upper/lower blade granule neuron distribution in $Sox9^{fl/fl};Nestin-Cre$. Because *Nestin-Cre* is mostly not active in the CH, this result may be explained by a contribution of CH-derived SOX9+ glial cells to formation of the supragranular scaffold, in further agreement of a mixed DNE/CH origin for this distal scaffold.

ALDH1L1, GLAST, BLBP, and GFAP co-expression in the fimbrial scaffold and its CH progenitors argue strongly for an astrocytic identity, demonstrating for the first time a migratory support role for this lineage in the mouse embryo. CH lineage tracing using $Wnt3a^{iresCre}$ further supports the astrocytic rather than multipotent radial glia nature of the scaffold because, following an early neurogenic phase, when CR cells are produced, CH progenitors then give rise to the GFAP+ scaffold, but not to granule neurons or OPCs. In the spinal cord, *Aldh1l1*-GFP is first detected at E12.5 (*Tien et al., 2012*) and we observe a similar onset of expression of ALDH1L1 in the CH astrocytic progenitors, which we suggest may represent the fimbrial glioepithelium (*Barry et al., 2008*). This is in contrast with what is observed elsewhere in the developing forebrain where astrocytes are known to arise around E16.5 (*Bayraktar et al., 2014*), and suggests an earlier emergence in this region. Interestingly, CH-derived CR cells are the first neurons generated in the brain (*Takiguchi-Hayashi et al.,*

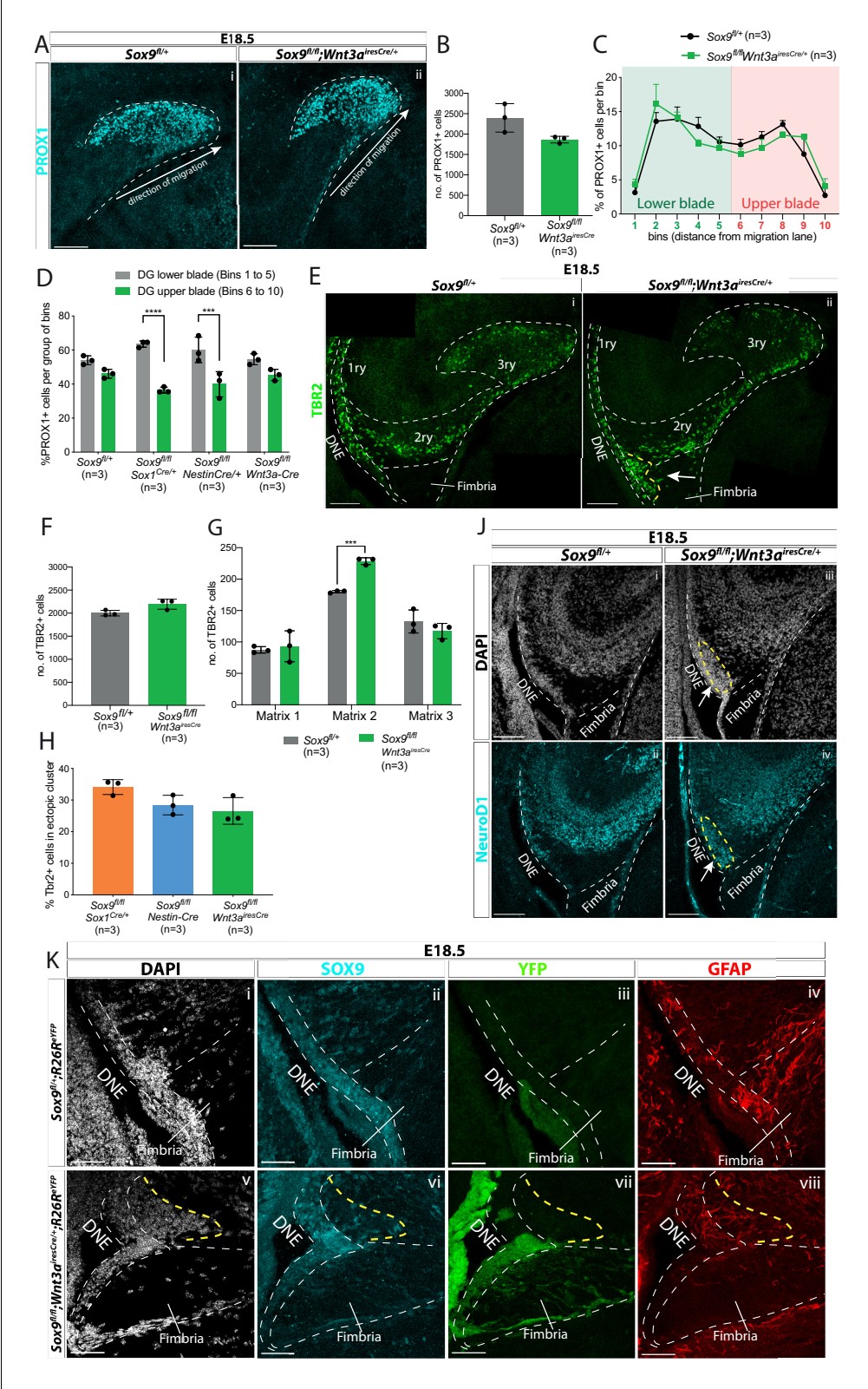

**Figure 7.** CH-specific deletion of *Sox9* using *Wnt3a^iresCre^* specifically affects granule neuron progenitor migration along the 1ry-to-3ry matrix axis. (A–D) Analysis of PROX1+ differentiating granule neurons in E18.5 *Sox9^fl/fl^;Wnt3a^iresCre/+^* dentate gyrus (DG). (A) Immunostaining for PROX1 on E18.5 controls and *Sox9^fl/fl^;Wnt3a^iresCre/+^* brains. The total number of PROX1+ cells (B) and their distribution within the forming DG (C, see *Figure 3N* for analysis settings) was not affected in *Sox9^fl/fl^;Wnt3a^iresCre/+^* mutants compared to controls. (D) Percentage of PROX1+ granule neurons positioned in the DG

*Figure 7 continued on next page*

*Figure 7 continued*

lower blade (bins 1–5), versus the DG upper blade (bins 6–10) in E18.5 controls, *Sox9$^{fl/fl}$;Sox1$^{Cre/+}$* and *Sox9$^{fl/fl}$;Nestin-Cre* mutants (results from **Figure 2N**), and *Sox9$^{fl/fl}$;Wnt3a$^{iresCre/+}$* mutants (results from **C**). In contrast with *Sox9$^{fl/fl}$;Sox1$^{Cre/+}$* (bins 1–5: 63.53 ± 1.85%, bins 6–10: 36.43 ± 1.70%, p=<0.0001) and *Sox9$^{fl/fl}$;Nestin-Cre* mutants (bins 1– 5: 60.10 ± 7.47%, bins 6–10: 39.87 ± 7.43%, p=0.0002), PROX1+ granule neurons distribution in *Sox9$^{fl/fl}$;Wnt3a$^{iresCre/+}$* mutants (bins 1–5: 54.60 ± 3.26%, bins 6–10: 45.35 ± 3.30%) is similar to controls (bins 1–5: 54.00 ± 2.61%, bins 6–10: 46.03 ± 2.60%; Sidak multiple comparison test, Two-way ANOVA interaction p=0.0044). (**E–H**) Analysis of TBR2+ intermediate progenitors at E18.5 in *Sox9$^{fl/fl}$; Wnt3a$^{iresCre/+}$* DG via immunofluorescence (**E**). The total number of TBR2+ cells is unchanged (**F**) but their distribution along the three matrices (**G**) is affected as there were more cells in the 2ry matrix of *Sox9$^{fl/fl}$;Wnt3a$^{iresCre/+}$* mutants (228.60 ± 5.37) compared to controls (180.07 ± 1.79, p=0.0001, *t* test). Arrow indicates accumulation of TBR2+ cells in the ectopic cluster in *Sox9$^{fl/fl}$;Wnt3a$^{iresCre/+}$* mutants. (**H**) Percentage of TBR2+ in ectopic cluster. The percentage of TBR2+ cells in the ectopic cluster is comparable to that observed *Sox9$^{fl/fl}$;Sox1$^{Cre/+}$* and *Sox9$^{fl/fl}$;Nestin-Cre* mutants (calculated as % of TBR2+ progenitors in ectopic cluster relative to total number of TBR2 progenitors in 2ry matrix). (**J**) Immunofluorescence for NeuroD1 showing ectopic differentiation toward granule neuron cell fate in the ectopic cluster of *Sox9$^{fl/fl}$;Wnt3a$^{iresCre/+}$* mutants (arrow). (**K**) Triple immunostaining for SOX9, YFP, and GFAP on E18.5 controls and *Sox9$^{fl/fl}$;Wnt3a$^{iresCre/+}$* brains showing YFP- cells accumulating next to the SOX9+ DNE in E18.5 *Sox9$^{fl/fl}$; Wnt3a$^{iresCre/+}$;R26R$^{eYFP}$* mutants (delineated by yellow dashed line) and underlaid by a defective GFAP scaffold. DNE: dentate neuroepithelium. Scale bars represent 50 µm in (**K**); 100 µm in (**A**), (**E**), and (**J**).

The online version of this article includes the following source data and figure supplement(s) for figure 7:

**Source data 1.** Quantification of total number and distribution of TBR2 and PROX1-expressing cells and ectopic matrix size during dentate gyrus (DG) development in *Sox9$^{fl/fl}$;Wnt3a$^{iresCre/+}$* mutants compared to controls.

**Figure supplement 1.** Histological analysis of *Sox9$^{fl/fl}$;Wnt3a$^{iresCre/+}$* E18.5 developing DG.

**2004**). Since the gliogenic switch is also controlled by differentiating neurons (**Barnabé-Heider et al., 2005**), it is tempting to speculate that the initial emergence of CR cells may explain early gliogenic induction in the CH. Finally, in the supragranular scaffold, ALDH1L1 and GFAP are both present but they do not colocalise. This supports a mixed origin for this part of the scaffold, and furthermore suggests a different cellular composition. In fact, exclusive expression of GFAP in some cells suggests that, in this domain, DNE-derived RGCs may support neuronal migration, in addition to some CH-derived astrocytic progenitors.

Other migration cues are necessary for granule neuron guidance. CH-derived CR cells play an important role through the release of chemokines, such as Reelin (**Frotscher et al., 2003**) and SDF1 (CXCL12) (**Berger et al., 2007**). We did not observe any significant alteration in *Reelin* and *Cxcl12* expression indicating that CR cells are not affected by *Sox9* deletion. Therefore, our work highlights a crucial role for the fimbrial glial scaffold for 1ry-to-3ry matrix progenitor migration from E18.5, whereas the supragranular scaffold facilitate 3ry matrix cell distribution (Figure 8E,F). Detailed investigations are needed to characterize how the fimbrial glial scaffold interact with progenitors and support their migration and/or delamination from the DNE.

## Role of SOX9 during DG morphogenesis

Impairment of DNE progenitor migration in *Sox9$^{fl/fl}$;Wnt3a$^{iresCre/+}$* mutants demonstrates that loss of SOX9 in these cells does not explain the migration defect. However, this does not rule out a role for SOX9 in DNE progenitors. Indeed, while we did not observe a significant alteration in progenitor emergence, differentiation, and survival embryonically, there is a significant decrease in IP numbers early post-natally. This reduction coincides with formation of the ectopic cluster, where 35% of TBR2 + progenitors are unable to reach the developing DG in *Sox9$^{fl/fl}$;Sox1$^{Cre/+}$* mutants (**Figure 3Ei**). Because their survival and expansion might be affected due to their ectopic location, this might account for the reduction in total progenitor numbers. While we did not detect a significant increase in apoptosis in the ectopic cluster at P2, these cells are absent in the adult brain, suggesting a post-natal loss. Further analyses are required to understand their fate, for example, whether their migration resumes, and whether this follows the scaffold or not. Alternatively, reduction of TBR2+ cells could indicate that SOX9 is required for maintenance and/or expansion of migrating progenitors and NSCs (**Nelson et al., 2020**). Loss of *Sox9* in these cells could furthermore lead to postnatal reduction of a pool of newly formed TBR2 cells that was not detected by our analysis. The milder phenotype in *Sox9$^{fl/fl}$;Nestin-Cre* embryos could thus be explained by a cell autonomous requirement for SOX9 in DNE cells. Additionally, early and transient expression of SOX9 in these mutants could underlay the lesser defects.

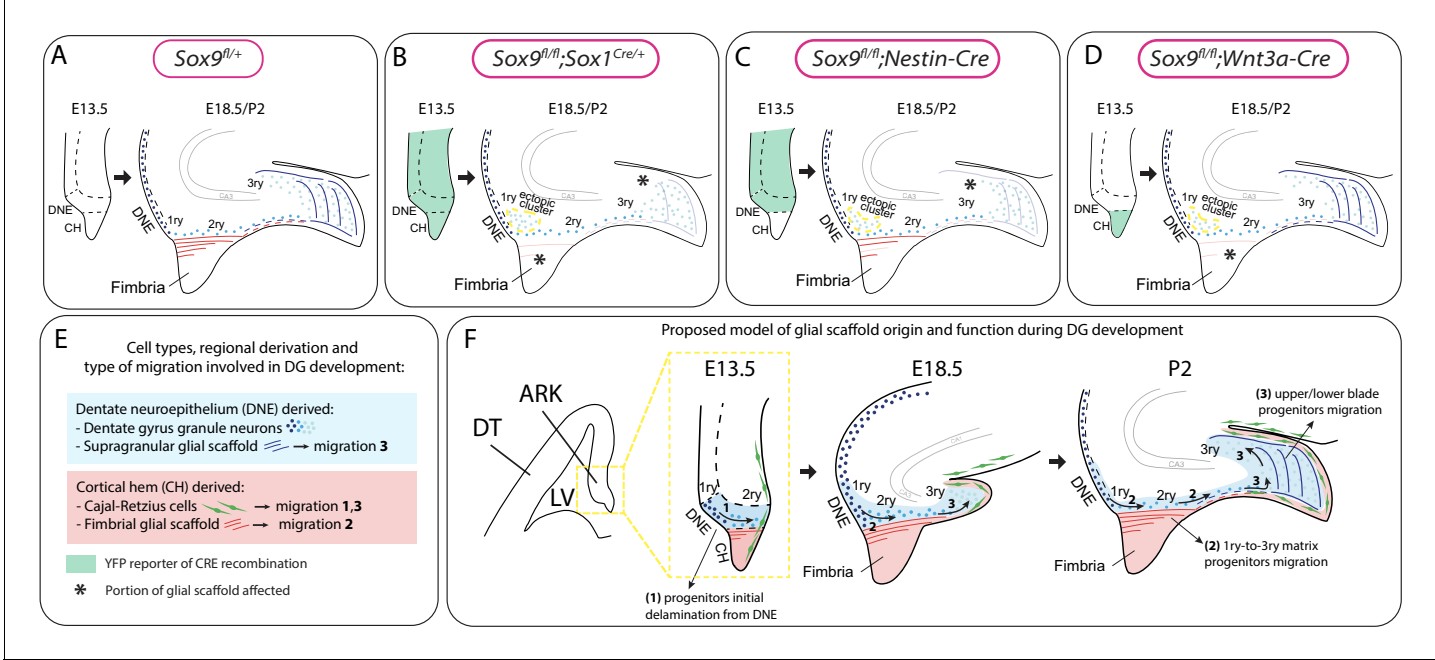

**Figure 8.** Model for the dual origin and function of the dentate gyrus (DG) glial scaffold based on the analysis of defects following differential deletion of *Sox9*. (A–D) Schematic of mouse models used for *Sox9* conditional deletion and analysis of DG development. The pattern of Cre recombination is represented in the archicortex at E13.5 (green area) and the corresponding phenotype observed at E18.5/P2. Stars indicate local absence of the GFAP+ glial scaffold. (E) Figure legend. (F) Model of DG development based on defects observed following differential deletion of *Sox9*. At early stages of DG development (E13.5), granule neuron progenitors undergo delamination from the 1ry matrix and form the 2ry matrix (migration direction depicted by arrow 1). This initial step is hypothesized to be independent of the glial scaffold because it is not affected in its absence in *Sox9* mutants. From E18.5, progenitor migration toward the forming DG/3ry matrix relies on the fimbrial scaffold (red lines, arrow 2). This fimbrial scaffold derives from astrocytic progenitors located in the CH (red area). At the same time, the dentate scaffold around the DG (blue lines) provides support for granule neuron positioning within upper and lower blades of the forming DG (arrows 3). Cells giving rise to this second scaffold are DNE derived (blue area). CH: cortical hem; DNE: dentate neuroepithelium; HNE: hippocampal neuroepithelium; VZ: ventricular zone; DT: dorsal telencephalon; ARK: archicortex.

In support of a role of SOX9 in the DNE, adult DG functionality is affected by embryonic deletion of SOX9, because we observe compromised memory formation abilities in *Sox9^{fl/fl};Sox1^{Cre/+}* adults. This is the most straightforward explanation; however, we cannot exclude that other brain regions potentially affected by *Sox9* deletion may also impact on this impaired behavior (e.g. altered loco-motion, smell or sight). We (*Scott et al., 2010*) and others *Hashimoto et al., 2016*; *Güven et al., 2020* have indeed previously shown that SOX9 regulates progenitor formation and expansion. However, in other contexts, such a role was not observed (*Kang et al., 2012*; *Stolt et al., 2003*; *Vong et al., 2015*; *Martini et al., 2013*). This variability may depend on the cellular context, but timing of the deletion is also relevant. Compensation by other members of the SOXE family, in particular SOX8, has been shown to explain recovery of some defects due to *Sox9* loss (*Weider and Wegner, 2017*). In our context, it is likely that both compensatory mechanisms and timing contribute to the difference in the severity of the defects observed after *Sox9* loss in different models. Analysis of SOX9/SOX8 double mutants would clarify this possibility. Finally, SOX9 is also expressed in adult DG NSCs (*Shin et al., 2015*), where it might be required for their maintenance, as shown for SVZ NSCs (*Scott et al., 2010*). As discussed above, compromised memory-forming abilities are observed in *Sox9^{fl/fl};Sox1^{Cre/+}* adults, which could well reflect the reduced numbers of neuronal progenitors reaching the DG. However, impaired adult neurogenesis could contribute to this phenotype, and although it is beyond the scope of the current study, this warrants further investigation. Moreover, as discussed above, SOX9 is likely to be required in DNE progenitors for formation of the supragranular scaffold, most likely as an inducer of gliogenic or RGCs fate (*Scott et al., 2010*; *Stolt et al., 2003*).

The importance of SOX9 for the acquisition of gliogenic potential is demonstrated by the loss of the fimbrial scaffold in *Sox9^{fl/fl};Wnt3a^{iresCre/+}* mutants. In the astrocytic lineage, SOX9 expression is

maintained at high levels (*Sun et al., 2017*) and it is required for astrocytic specification in the spinal cord (*Kang et al., 2012*) and anterior CNS (*Scott et al., 2010*; *Nagao et al., 2016*; *Güven et al., 2020*). In the spinal cord it has been shown to induce expression of NFIA, with which it then interacts to activate expression of astrocytic genes (*Kang et al., 2012*). In contrast, we show here that NF1A/B expression is not affected in the developing forebrain following *Sox9* deletion. Therefore other factors and pathways must be involved for induction of *Nf1a* expression, which could include BRN2 (*Glasgow et al., 2017*), WNT (*Hasenpusch-Theil et al., 2012*), and/or NOTCH (*Namihira et al., 2009*). Furthermore, as shown in the spinal cord (*Kang et al., 2012*), gliogenesis in CH is simply delayed in *Sox9^{fl/fl}*;*Sox1^{Cre/+}* and *Sox9^{fl/fl}*;*Nestin-Cre* mutants. This could reflect a reduced transcriptional activity of NF1 factors without SOX9, as previously shown for NF1A (*Kang et al., 2012*). As discussed above, recovery could also be due to compensatory expression of SOX8. Further analyses are required to understand the mechanisms underlying recovery of the fimbrial scaffold.

### Conclusions

The ability of the fimbrial glial scaffold to support 1ry-to-3ry matrix progenitor migration is an exciting new finding. It will be important to characterize the molecular mechanism utilized by the scaffold for this function, in fact whether this support is based on release of chemoattractants and/or relies on cell-cell or cell-matrix interactions, is still unknown. We show that astrocytes forming the fimbrial scaffold are closely intermingled with migrating neuronal progenitors of the 2ry matrix, in fact they could form tubules around them similarly to how this same cell type supports neuroblast migration along the rostral migratory stream of the adult brain (*Lois et al., 1996*; *Gengatharan et al., 2016*). This aspect of scaffold functionality could be addressed using *Aldh1l1-Cre* (*Tien et al., 2012*) to delete ligands potentially involved.

In addition, our CH lineage tracing analysis using *Wnt3a^{iresCre}*;*R26R^{eYFP}* shows that progenitors in the CH exclusively generate CR cells and then switch to formation of the glial scaffold. However, we found around 6.5% of granule neurons in the *Wnt3a^{iresCre}*;*R26R^{eYFP}* lineage, suggesting that either *Wnt3a^{iresCre}* is ectopically active in the DNE, or that a proportion of neuronal progenitors may originate from the CH. This latter exciting possibility requires further confirmation with additional lineage tracing analyses; it also suggests that this subpopulation may have different characteristics. Whether other cells forming the adult DG, such as adult NSCs, also originate from the CH, is unknown and requires further investigation.

In conclusion, SOX9 plays sequential roles during CNS development as cells progress from a NSC fate, in which the protein is required for induction and also maintenance, to acquisition of gliogenic potential. Experimental manipulation of its expression levels highlights aspects of its function, according to the cellular context and also timing, which is presumably explained by the pattern of expression of redundant SOXE members, and its different interactors. Here, we reveal that DG development is particularly sensitive to early loss of *Sox9* and that this is at least in part due to the failure to generate an astrocytic scaffold that aids neuronal migration. Extensive cell migration (*Treves et al., 2008*) and progenitor pool expansion (*Martin et al., 2002*) might underlay the vulnerability of this region, illustrated by the lasting consequences of transiently impaired cell migration.

## Materials and methods

### Key resources table

| Reagent type (species) or resource | Designation | Source or reference | Identifiers | Additional information |
|---|---|---|---|---|
| Genetic reagent (*Mus musculus*) | *Sox9^{fl/fl}* | *Akiyama et al., 2002* | *Sox9^{tm2Crm}* MGI: 2429649 | Conditional targeted mutation |
| Genetic reagent (*M. musculus*) | *Sox1^{Cre/+}* | *Takashima et al., 2007* | *Sox1^{tm1(cre)Take}* MGI: 3807952 | Targeted mutation |
| Genetic reagent (*M. musculus*) | *Nestin-Cre* | *Tronche et al., 1999* | *(no gene)^{Tg(Nescre)1Kln}* MGI: 2176173 | Transgenic insertion |
| Genetic reagent (*M. musculus*) | *Wnt3a^{iresCre}* | *Yoshida et al., 2006* - | *Wnt3a^{tm1.1(cre)Mull}* MGI: 98956 | Targeted mutation |

*Continued on next page*

| Reagent type (species) or resource | Designation | Source or reference | Identifiers | Additional information |
|---|---|---|---|---|
| Genetic reagent (*M. musculus*) | R26R^eYFP | *Srinivas et al., 2001* | Gt(ROSA)26Sor^tm1(EYFP)Cos MGI: 2449038 | Targeted mutation |
| Antibody | Anti- ALDH1L1 (rabbit polyclonal) | Abcam | Cat# ab87117, RRID: AB_10712968 | IF (1:500) |
| Antibody | Anti-BLBP (rabbit polyclonal) | Millipore | Cat# ABN14, RRID: AB_10000325 | IF (1:200) |
| Antibody | Anti-Caspase (rabbit polyclonal) | R and D system | Cat# AF835, RRID: AB_2243952 | IF (1:400) |
| Antibody | Anti-GFAP-Cy3 (mouse monoclonal) | Sigma | Cat# C9205, RRID: AB_476889 | IF (1:500) |
| Antibody | Anti-GLAST (guinea pig polyclonal) | Millipore | Cat# AB1782, RRID: AB_90959 | IF (1:200) |
| Antibody | Anti-LEF1 (rabbit polyclonal) | Cell Signalling | Cat# 2230, RRID: AB_823558 | IF (1:200) |
| Antibody | Anti-NF1A (rabbit polyclonal) | Active Motif | Cat# 39397, RRID: AB_2314931 | IF (1:500) |
| Antibody | Anti-NF1B (rabbit polyclonal) | Abcam | Cat# ab186738, RRID: AB_2782951 | IF (1:200) |
| Antibody | Anti-PAX6 (rabbit polyclonal) | Covance | Cat# PRB-278P, RRID: AB_291612 | IF (1:300) |
| Antibody | Anti-PROX1 (rabbit polyclonal) | BioLegend | Cat# PRB-238C, RRID: AB_291595 | IF (1:500) |
| Antibody | Anti-REELIN (mouse monoclonal) | Abcam | Cat# ab78540, RRID: AB_1603148 | IF (1:200) |
| Antibody | Anti-SOX2 (goat polyclonal) | Neuromics | Cat# GT15098, RRID: AB_2195800 | IF (1:500) |
| Antibody | Anti-SOX9 (goat polyclonal) | R and D system | Cat# AF3075, RRID: AB_2194160 | IF (1:200) |
| Antibody | Anti-TBR2 (rabbit polyclonal) | Abcam | Cat# ab23345, RRID: AB_778267 | IF (1:500) |
| Antibody | Anti-GFP (rat monoclonal) | Fine chemical products | Cat# 04404–84, RRID: AB_10013361 | IF (1:1000) |
| Recombinant DNA reagent | pCAG-hyPBase (plasmid) | *Mikuni et al., 2016* | | Plasmids for in utero electroporation (1 µg/µl) |
| Recombinant DNA reagent | pPB-CAG-DsRed (plasmid) | *Mikuni et al., 2016* | | Plasmids for in utero electroporation (1 µg/µl) |
| Software, algorithm | Ethovision XT | Noldus | RRID:SCR_000441 | |
| Software, algorithm | Distance.gui | This paper | | Source code file provided (see Source Code File 1) |

## Mouse strains, husbandry, and genotyping

All experiments carried out on mice were approved under the UK Animal (scientific procedures) Act 1986 (Project license n. 80/2405 and PP8826065). Mouse husbandry, breeding, ear biopsies and vaginal plug (VP) checks were performed by the Biological Research Facility team of the Francis Crick Institute. Animals were kept in individually ventilated cages (ICV) with access to food and water ad libitum. The VP day was considered as 0.5 day from time of conception (E0.5) and the day of birth termed P0.

All mouse lines used were previously described: *Sox9^fl/fl* conditional targeted mutation, MGI: 2429649 (*Akiyama et al., 2002*); *Sox1^Cre/+* targeted mutation, MGI: 3807952 (*Takashima et al., 2007*); *Nestin-Cre* transgenic mutation, MGI: 2176173 (*Tronche et al., 1999*); *Wnt3a^iresCre* targeted mutation, MGI: 98956 (*Yoshida et al., 2006*); *R26R^eYFP* targeted mutation, MGI: 2449038 (*Srinivas et al., 2001*). To obtain *Sox9* conditional mutations, *Sox9^fl/fl* mice were crossed with either *Sox1^Cre/+*, *Nestin-Cre* or *Wnt3a^iresCre* mice. All Cre lines were kept in heterozygosity. To verify Cre

recombination pattern, *R26R^{eYFP}* reporter allele was also present in all samples analyzed, even when not indicated. Genotyping of embryos and adult mice was performed by Transnetyx.

## Behavioral analysis

The novel object recognition test (NORT) was used to analyze memory formation in adult mice, as the ability to discern between new and familiar objects. A 40 × 40 cm arena made of white Plexiglas was built by the Francis Crick Institute mechanical engineering facility. Pairs of different objects were switch between cohorts of animals to avoid biases due to object conformation. Logitech 910C webcam and software were used to record behavioral tests and videos were used for analyses. Mice were acclimatized to the testing room for 1 hr before starting the test. The operator was alone with the mice during the duration of the test to avoid any disturbance. The behavioral test was performed on 3 consecutive days, and each mouse spends 5 min in the arena per day. On the first day, mice were placed in an empty arena for adaptation. On the second day, mice were exposed to two identical objects, for training. On the third final day, mice were confronted to one familiar object (used the previous day) and a new, different object. Arena and objects were disinfected and rinsed before every run. Objects location within the arena was consistent among animals. The recordings from the first day of NORT (adaptation day) were used to perform open field test. Ethovision XT (Noldus) was used for the analysis of video recordings. For NORT, in each video, a circular area of 4 cm radius around each object was considered as object 'exploration area'. Time spent in this area was considered as 'exploration time'. Mice that did not explore both objects on the second day were excluded from further analysis. For the open-field test, the arena center was considered as a central square area 4 cm apart from the arena borders. Time spent in this area was considered as 'time spend it center' and was calculated with Ethovision XT.

## EdU injection

For cell birth-dating experiments, 10 mg/ml EdU solution, from Click-iT EdU imaging kit, was injected intraperitoneally in pregnant females at a dose of 30 µg/g body weight. Injection was performed at either E16.5 or E18.5 stage of pregnancy and samples collected at E18.5 or P2, respectively. Schematic of protocols in Figure S.8.E,I.

## Tissue harvesting and staining

Pregnant females were killed by cervical dislocation, embryonic heads were dissected out in chilled PBS and fixed by immersion in chilled 4% PFA at 4°C for 1–2 hr. From E16.5 onwards, brains were dissected out of the skull before fixation. P2 pups were killed by cervical dislocation, brains were dissected out on chilled PBS and fixed by immersion in chilled 4% PFA at 4°C for 2 hr. After fixation, embryonic brains were washed once in PBS, cryopreserved in sucrose 30%, then embedded in OCT, frozen on dry ice and stored at −80°C. Samples were cryosectioned at 14 µm and sections placed on Superfrost Plus glass slides, air dried for 5 min, washed twice in PBS for 5 min. For some antibodies (listed in *Table 1*), antigen retrieval was performed immerging slides in 10% target retrieval solution pH6.1 diluted 1:10 in distilled water, for 30 min at 65°C or 15 min at 95°C, then washing twice in PBS for 10 min. Slides were then incubated in a humidified chamber in blocking solution (10% donkey serum in 0.1% triton X-100 PBS) for at least 30 min, then incubated overnight at 4°C with primary antibodies diluted in blocking solution (dilution indicated in *Table 1*). The following day, sections were washed twice for 5 min in 0.1% triton X-100 PBS, incubated with secondary antibodies (*Table 2*) and DAPI diluted in blocking solution for 2 hr at room temperature in a dark humified chamber. Finally, sections were washed again twice for 5 min in PBS, briefly in distilled water, air dried and coverslip mounted with Aqua-poly/Mount. Apoptosis detection with terminal deoxynucleotidyl transferase dUTP nick end labeling (TUNEL) assay was performed using the ApopTag Red In Situ Apoptosis Detection Kit, following manufacturer's instructions, after secondary antibody incubation. Detection of DNA-incorporated EdU was performed using Click-iT EdU imaging kit, following manufacturer's instructions, after secondary antibody incubation.

For 3D reconstruction, samples were cryosectioned at 50 µm and sections placed floating in a 24-well plate in PBS. Sections were washed, processed for antigen retrieval at 65°C, and incubated in blocking solution (10% donkey serum in 0.5% triton X-100 PBS) as described above. Primary and secondary antibody incubation (antibodies diluted in 10% donkey serum in 0.5% triton X-100 PBS;

**Table 1.** List of primary antibodies used.
Antigen retrieval protocol (30 min in 65°C water bath or 15 min in 95°C decloaking chamber) was performed for the indicated samples (e: embryos; p: pups).

| Antigen | Host | Dilution | Vendor | Catalog # | 65°C | 95°C |
|---|---|---|---|---|---|---|
| ALDH1L1 | Rabbit | 1:500 | Abcam | ab87117 | e | |
| BLBP | Rabbit | 1:200 | Millipore | ABN14 | | |
| Caspase | Rabbit | 1:400 | R and D system | AF835 | | |
| GFAP-Cy3 | Mouse | 1:500 | Sigma | C9205 | | |
| GLAST | Guinea pig | 1:200 | Millipore | AB1782 | | |
| LEF1 | Rabbit | 1:200 | Cell Signalling | 2230P | | e |
| NF1A | Rabbit | 1:500 | Active Motif | 39397 | e | |
| NF1B | Rabbit | 1:200 | Abcam | ab186738 | e | |
| PAX6 | Rabbit | 1:300 | Covance | PRB-278P | | e |
| PROX1 | Rabbit | 1:500 | BioLegend | PRB-238C | e, p | |
| REELIN | Mouse | 1:200 | Abcam | ab78540 | e | |
| SOX2 | Goat | 1:500 | Neuromics | GT15098 | e | p |
| SOX9 | Goat | 1:200 | R and D system | AF3075 | e | |
| TBR2 | Rabbit | 1:500 | Abcam | ab23345 | | e, p |
| YFP | Rat | 1:1000 | Fine chemical products | 04404–84 | | |

dilution indicated in *Tables 1* and *2*) was 3 and 1 nights, respectively. Floating sections were then mounted on Superfrost Plus glass slides, air dried and coverslip mounted with Aqua-poly/Mount.

For hematoxylin and eosin (H and E) staining, pregnant females and adult mice were killed by cervical dislocation, embryonic and adult brains were dissected out in chilled PBS, fixed overnight in Bouin's solution at 4°C, washed twice for 10 min in 70% ethanol and stored in 70% ethanol until processing. Samples were embedded in wax, sectioned at 4 μm and stained by the Francis Crick Institute Experimental Histopathology facility.

### *In situ* hybridization probe formation and staining

Digoxigenin (DIG)-tagged antisense RNA probes were made from an ampicillin-resistant plasmid (kindly gifted by Dr. Paul Sharp) containing *Sox9* cDNA followed by a T7 promoter. Plasmid was amplified with *E. coli* culture and purified with NucleoBond Xtra Midi plus kit. Five μl of plasmid (corresponding to 5–10 μg) was linearized with 1 μl of SmaI enzyme, 2 μl of 10x SmartCut buffer and 12 μl of RNase-free water and confirmed with 1% agarose gel electrophoresis. Linearized plasmids were purified by phenol-chloroform extraction and precipitated by adding 1 μl glycogen, 1/20 of sodium acetate 3M and equal volume of 100% ethanol, incubated at −20°C for 1 hr. The precipitate was recovered by centrifugation at 13,000 RPM at 4°C for 15 min, air dried and resuspended in 16 μl

**Table 2.** List of secondary antibodies and nuclear staining used.

| Fluorophore | Host/reactivity species | Dilution | Vendor | Catalog# |
|---|---|---|---|---|
| Alexa 568 | Donkey anti-Rabbit | 1:500 | Thermo Fisher Scientific | A10042 |
| Alexa 647 | Donkey anti-Rabbit | 1:500 | Thermo Fisher Scientific | A31573 |
| Alexa 568 | Donkey anti-Goat | 1:500 | Thermo Fisher Scientific | A11057 |
| Alexa 647 | Donkey anti-Goat | 1:500 | Thermo Fisher Scientific | A21447 |
| Alexa 594 | Donkey anti-Mouse | 1:500 | Thermo Fisher Scientific | A21203 |
| Alexa 555 | Donkey anti-Mouse | 1:500 | Thermo Fisher Scientific | A31570 |
| Alexa 488 | Donkey anti-Rat | 1:500 | Thermo Fisher Scientific | A21208 |
| DAPI 300 μM | | 1:500 | Thermo Fisher Scientific | D1306 |

of RNase-free water. For DIG-tagged probes synthesis, 1 µl linearized plasmid, 1X transcription buffer, 2 µl of DIG-tagged nucleotides, 1 µl T7 RNA polymerase, 0.5 µl RNase inhibitor, 1 µl DTT 100 mM, and 11.5 µl of RNase-free water were incubated at 37°C for 2 hr. Probe formation was confirmed with 1% agarose gel electrophoresis. Probes were precipitated by adding 1 µl glycogen, 8 µl lithium chloride 5M, 2/3 of final volume of 100% ethanol, and 1/3 of final volume of RNase-free water, incubated at −80°C for 30 min. Precipitates were recovered by centrifugation at 13,000 RPM at 4°C for 15 min, then washed in 70% ethanol (v/v) centrifuged at 13,000 RPM for 15 min at 4°C. Pellet containing the RNA probes were then air-dried at 37°C and resuspended in 50–100 µl of hybridization buffer (50% (v/v) deionized formamide, 4X SSC, 0.01M β-mercaptoethanol, 10% dextran sulphate, 2X Denhart's solution, 0.23 mg/ml yeast t-RNA diluted in RNase-free water).

For *in situ* hybridization (ISH) staining, cryosections were initially air-dried, washed twice for 5 min in PBS, fixed 30 min with 4% PFA at room temperature in a humidified chamber, washed twice for 10 min in PBS, and incubated in pre-hybridization buffer (50% (v/v) deionized formamide, 1X saline-sodium citrate (SSC) diluted in RNase-free water) for 1 hr in a 65°C waterbath. For each slide, 1 µl of RNA probe was denatured in 200 µl hybridization buffer for 10 min at 70°C, then applied on sections. Hybridization was carried out in a humidified chamber overnight in a 65°C water bath. The following day, sections were washed twice for 15 min, and once for 30 min with pre-warmed washing buffer (50% (v/v) deionized formamide, 1X SSC, 0.1% (v/v) Tween 20 diluted in MilliQ water) at 65°C, then twice for 30 min with MABT (20 mM Maleic acid, 30 mM sodium chloride (NaCl), 0.02% (v/v) Tween 20, adjusted to pH7.5 and diluted in MilliQ water) at room temperature. Sections were then incubated in a humidified chamber at room temperature for 1 hr in blocking buffer (10% (v/v) sheep serum, 2% (v/v) blocking reagent diluted in MABT), then overnight in anti-DIG coupled to alkaline phosphatase antibody (α-DIG-AP) diluted 1:1500 in blocking buffer. The following day, sections were washed four times for 20 min at room temperature with MABT then twice for 10 min in pre-staining solution (100 mM NaCl, 50 mM magnesium chloride (MgCl2), 100 mM Tris-HCl pH9.5, 1% (v/v) Tween 20 in MilliQ water). Alkaline phosphatase staining was then performed incubating slides with staining solution (pre-staining solution plus 5% (w/v) Polyvinyl alcohol (PVA), 4.5 µl of Nitrotetrazolium Blue chloride (NBT), and 3.5 µl of 5-bromo-4-chloro-3-indolyl-phosphate (BCIP) per ml of staining solution) for 1 hr to 2 days at 37°C in a dark chamber. Once the desired staining intensity was reached, the reaction was stopped washing sections twice for 10 min in 0.1% triton X-100 PBS. Finally, sections were fixed for 10 min in 4% PFA, washed twice for 5 min in PBS then briefly in distilled water, air-dried, and coverslip mounted with Aqua-poly/Mount.

## Quantitative PCR and analysis

For gene expression analysis, E12.5 embryos were dissected in sterile PBS, dorsal telencephalon and archicortex were separately snap-frozen in liquid nitrogen. RNA was extracted with the RNeasy Plus Micro kit, following manufacturer's instructions. Complimentary (c) DNA was synthetized from 250 ng of extracted RNA in 1X qScript cDNA SuperMix diluted in RNase-free water and incubated on the Tetrad 2 Thermal Cycler following the indicated reverse transcription protocol. Resulting cDNA was diluted in RNase-free water at a final concentration of 200 µg/µl. Transcripts were quantified with quantitative PCR (qPCR), mixing 800 µg of cDNA in 1X ABsolute QPCR SYBR Green ROX Mix and 40 nM of primer mix (forward and reverse primers were pre-mixed; primers sequences are indicated in *Table 3*). Expression level of the house-keeping gene Glyceraldehyde 3-phosphate dehydrogenase (GAPDH) was used as reference. Each sample was run in technical triplicates, in case of high variability, one of the technical triplicates was removed. Number of biological replicates are indicated for each experiment (n).

Relative expression of the genes of interest were calculated by normalization of the detected expression value to the geometric mean of the reference GAPDH gene using the ΔΔCt method (*Livak and Schmittgen, 2001*). More precisely, average cycle threshold (Avg Ct) was first calculated among technical triplicates or duplicates of each sample. Average delta Ct (Avg ΔCt) was then deduced by subtracting GAPDH Avg Ct to sample Avg Ct. The relative quantification (RQ) of cDNA for each gene was calculated as $2^{-Avg\Delta Ct}$. The fold change of each sample was calculated in reference to the average RQ of control samples group (control RQ) as: sample RQ/control RQ. The qPCR final results are shown as histograms, where each bar shows the average fold change of experimental replicates. Error bars are represented as standard error of the mean (SEM).

**Table 3.** List of primers used for qPCR.

| Target | Forward primer (5'- . . . −3') | Reverse primer (5'- . . . −3') | Supplier and Catalog # |
|---|---|---|---|
| *Gapdh* | TTCACCACCATGGAGAAGGC | CCCTTTTGGCTCCACCCT | Eurofins |
| *Sox9* | AAGAAAGACCACCCCGATTACA | CAGCGCCTTGAAGATAGCATT | Eurofins |
| *Nf1a* | CTTTGTACATGCAGCAGGAC | TTCCTGCAGCTATTGGTGTTT | Eurofins |
| *Nf1b* | GTGGAACCGGTGAATCTTTC | TCTGTCCTGGGCTCTATTCC | Eurofins |
| *Aldh1l1* | N/A | N/A | Qiagen PPM27706B-200 |
| *Cxcr4* | N/A | N/A | Qiagen PPM03149E-200 |
| *Cxcl12* | TGCATCAGTGACGGTAAACCA | TTCTTCAGCCGTGCAACAATC | Eurofins |
| *Reln* | TTACTCGCACCTTGCTGAAAT | CAGTTGCTGGTAGGAGTCAAAG | Eurofins |
| *Vldlr* | GGCAGCAGGCAATGCAATG | GGGCTCGTCACTCCAGTCT | Eurofins |

### *In utero* electroporation

For *in utero* electroporation (IUE), the *piggyBac* transposon system was used to avoid episomal plasmid loss upon cell division. pCAG-hyPBase and pPB-CAG-DsRed plasmids were kindly donated by Dr. Lucas Baltussen and have been previously described (*Mikuni et al., 2016*). Plasmids was amplified with *E. coli* culture, purified with EndoFree Plasmid Maxi kit and mixed together at a concentration of 1 µg/µl per plasmid, with 0.05% of FastGreen in injectable water.

IUE was performed on E15.5 embryos. One before surgery, analgesia (Carprofen, dose 50 mg/ml) was administered via drinking water to single caged pregnant females. On surgery day, pregnant females were anesthetised using isoflurane and subcutaneous injection (10 mg/kg of meloxicam and 0.1 mg/kg of buprenorphine in injectable water). Females' eyes were kept moist using Viscotears eye gel. Anesthetized female was shaved on the abdomen, cleaned with chlorhexidine and moved to surgical area, where body temperature was monitored on a heating pad. Laparotomy and exteriorization of the uterus were then performed. Ten µl of DNA was loaded using micro loader tips into Ethylene Oxide (EtO) gas sterilized borosilicate glass capillaries (1.0 OD x 0.58 ID x 100 L mm) which were pulled with a micropipette puller and the tip was broken using forceps. One µl of solution containing the plasmid DNA was injected into the lateral ventricle of each embryos using a Femtojet pico dispenser, followed by electroporation (5 pulses 38V of 50 ms with 1 s interval) with EtO gas sterilized 5 mm paddle type electrodes. The uterus was gently reinserted into the abdomen, then abdominal wall and skin were sutured separately. Mice were placed in a recovery chamber for a few hours. Analgesia (Carprofen, dose 50 mg/ml) was administrated in drinking water for the following 48 hr. Electroporated embryos were harvested at P2.

### Software for cell migration analysis

Distance.gui software was used to analyse PROX1+ cells distribution within the forming DG. It was written by Dr. Vivien Labat-gest and kindly donated by Prof. Federico Luzzati. The software calculates the distance in pixels between a point and line, which in this case are single PROX1+ cells and their migration line, respectively (schematic in *Figure 2N*). Therefore, the software input files for each image are the ImageJ cell counter result *.xml* file (representing PROX1+ cells point coordinates) and the XY coordinates of the reference line extracted from ImageJ as a *.txt* file (representing the migration line). The output file is a *.txt* file containing a list of numbers representing the distance in pixel of each PROX1+ cell from the migration line. For each picture, the range of PROX1+ cell distribution (most distant cell from the migration line) was used to divide the forming DG area in 10 bins, then percentage of PROX1+ cell per bin was calculated and plotted as a line (*Figures 2N* and *7C*). Cells in bins #1–5 would be closer to the migration line, therefore representing the lower DG blade, compared to cells in bins #6–10, representing the upper DG blade.

## Statistical and image analysis

H and E and *ISH*-stained sections were acquired with Leica DM750 light microscope, and LAS (Leica Application Suite) EZ software was used for acquisition. Sections processed for EdU, TUNEL, and immunofluorescence staining were imaged using Leica TCS SPE confocal microscope with ×10, ×20, and ×40 objectives. LAS AF software was used for acquisition. Acquisitions were performed as 1.5 μm Z-stacks, with bidirectional X. For image analysis and counting, ImageJ and QuPath softwares were used. For cell quantification, five different images were acquired and counted per analysed area for each sample. GFAP immunofluorescence was quantified based on positive pixel area after setting a threshold, using ImageJ. For 3D reconstructions, acquisitions were performed as 1 μm Z-stacks with bidirectional X. IMARIS was used image processing.

Statistical analysis of cell number quantification and qPCR analysis was performed on Prism 7 (Graphpad), calculating student's two-sided unpaired *t* tests, when comparing two groups, or ordinary one-way ANOVA, when comparing one variable in three or more groups, or ordinary two-way ANOVA, when comparing two variables in three or more groups. When performing ANOVA, multiple comparison between each experimental group was then performed with Tukey's test or Sidak's test, respectively. Analyses were performed parametrically, upon confirmation of samples normal distribution (performed on Prism). When the majority of sample groups within one analysis are not normally distributed (two out of three), statistical analysis was performed using Kruskal-Wallis test. Full details of statistical analyses can be found in the source data.

Histograms represent average quantification from the indicated number of biological replicates (n, minimum 3). Error bars for cell number quantification represent standard deviation (SD). Error bars for qPCR fold change analysis represent standard error of the mean (SEM). Data shown as percentage were processed with angular transformation before statistical analysis. p Value is indicated as: ns:p>0.05; *p≤0.05; **p≤0.01; ***p≤0.001; ****p≤0.0001.

## Acknowledgements

We are grateful for help and support from all past and present members of the Lovell-Badge's lab. We thank François Guillemot (the Francis Crick Institute) for helpful discussions. Distance.gui software for analysis of granule neurons distribution within the forming DG was written by Dr. Olivier Pierre Friard and kindly donated by Prof. Federico Luzzati (Neuroscience Institute Cavalieri Ottolenghi – NICO). We also thank Dr. Lucas Baltussen for providing training for in utero electroporation and kind donation of pCAG-hyPBase and pPB-CAG-DsRed plasmids. We thank Dr Celia Garau and Dr Richard Lilley for assistance in setting up the behavioral test and analysis. Finally, we also thank Biological Services, Experimental Histopathology, Advanced light microscopy and Mechanical Engineering platforms at the Francis Crick Institute, for their excellent assistance and technical support and the Mutant Mouse Regional Resource Center U42OD010918 for providing aliquots of *Wnt3a*^ires-Cre^ cryo-preserved sperm. This work was supported by the Medical Research Council, U.K. (U117512772, U117562207 and U117570590) and the Francis Crick Institute which receives its core funding from Cancer Research UK (FC001107), the UK Medical Research Council (FC001107), and the Wellcome Trust (FC001107).

## Additional information

### Funding

| Funder | Grant reference number | Author |
| --- | --- | --- |
| Medical Research Council | U117512772 | Robin Lovell-Badge<br>Robin Lovell-Badge |
| Medical Research Council | U117562207 | Robin Lovell-Badge<br>Robin Lovell-Badge |
| Medical Research Council | U117570590 | Robin Lovell-Badge<br>Robin Lovell-Badge |
| Cancer Research UK | FC001107 | Robin Lovell-Badge<br>Robin Lovell-Badge |

| Medical Research Council | FC001107 | Robin Lovell-Badge |
| | | Robin Lovell-Badge |

The funders had no role in study design, data collection and interpretation, or the decision to submit the work for publication.

## Author contributions

Alessia Caramello, Data curation, Formal analysis, Investigation, Methodology, Writing - original draft, Writing - review and editing; Christophe Galichet, Methodology, Writing - review and editing; Karine Rizzoti, Conceptualization, Supervision, Writing - original draft, Project administration, Writing - review and editing; Robin Lovell-Badge, Conceptualization, Supervision, Funding acquisition, Project administration, Writing - review and editing

## Author ORCIDs

Alessia Caramello (ID) https://orcid.org/0000-0001-7455-7686
Christophe Galichet (ID) https://orcid.org/0000-0003-0244-9381
Karine Rizzoti (ID) https://orcid.org/0000-0003-0711-5452
Robin Lovell-Badge (ID) https://orcid.org/0000-0001-9364-4179

## Ethics

Animal experimentation: All experiments carried out on mice were approved under the UK Animal (scientific procedures) Act 1986 (Project license n. 80/2405 and PP8826065).

## Decision letter and Author response

Decision letter https://doi.org/10.7554/eLife.63904.sa1
Author response https://doi.org/10.7554/eLife.63904.sa2

# Additional files

## Supplementary files

• Source code 1. The Distance.gui software was used to analyse cell distribution as detailed in the Materials and method section. It was written by Dr. Vivien Labat-gest and kindly donated by Prof. Federico Luzzati.

• Transparent reporting form

## Data availability

All data generated or analysed during this study are included in the manuscript and supporting files. Source data files have been provided where required.

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
