## [Decision Letter]

**Acceptance summary:**

The manuscript considered the reviewer comments carefully, and addressed the weaknesses with new data to demonstrate more conclusively that SOX9-dependent fimbria glial scaffold guides the migration of granule neuron progenitors towards dentate gyrus, and how formation of a local network, amidst astrocytic and neuronal progenitors originating from adjacent domains, underlays brain morphogenesis. Please take note of the points below and we hope you will continue to support *eLife*.

**Decision letter after peer review:**

Thank you for submitting your article "Dentate gyrus development requires a cortical hem-derived astrocytic scaffold" for consideration by *eLife*. Your article has been reviewed by three peer reviewers, and the evaluation has been overseen by a Reviewing Editor and Marianne Bronner as the Senior Editor. The following individuals involved in review of your submission have agreed to reveal their identity: Michael Piper (Reviewer #2); Robert F Hevner (Reviewer #3).

The reviewers have discussed the reviews with one another and the Reviewing Editor has drafted this decision to help you prepare a revised submission.

Summary:

The study by Caramello et al. demonstrates the importance of the fimbrial-specific glial scaffold in the migration of granule neuron progenitors towards dentate gyrus, and that the formation of such scaffold depends on SOX9. Using different Cre drivers, the authors reported solid phenotypes in the *Sox9* mutants, including the ectopic cluster of neuronal progenitors outside DNE and the impaired glial scaffold formation at the fimbria. The role of fimbria glia scaffold in the migration of DG neuronal progenitors was suggested previously but without concrete evidence. It is an important advance to show that the glial scaffold arises from both CH and DNE origins, with preferential localization of each along the fimbrial and distal DG ("supragranular glial scaffold") regions, respectively. Perhaps, there is something about the CH/DNE boundary that induces glial fiber differentiation. Quantification and statistics are used appropriately throughout and increases confidence. Therefore, the findings here are interesting and will be significant to the field. However, the proposed model lacks direct evidence and further experiments will be required to strengthen the current conclusions.

1) The primary finding of the manuscript is that SOX9-dependent fimbria glial scaffold guides the migration of granule neuron progenitors towards dentate gyrus. This is mainly supported by the phenotypic difference between *Sox9* knockout driven by *Sox1-Cre* and *Nestin-Cre*, with the former enables *Sox9* deletion at the cortical hem while the latter doesn't (Figure 1G). This makes proving the *Nestin-Cre* mutants have mild or no phenotypes particularly important. In several cases, the images actually suggest *Nestin-Cre Sox9* mutants do have a phenotype, but the statistical analysis indicate no significant difference (e.g. Figure 1A, the size of DG looks smaller but the graph indicates not; Figure 2G, no. of TBR2+ cells in the graph is reduced; Figure 4C, GFAP expression at E18.5 seems reduced in the graph). The authors should increase the sample size to provide more solid and consistent results. It would be useful to quantify GFAP fluorescence intensity in panel C, as was done in Figure 4.

2) Figure 1: Do the *Sox9*/*Sox1cre* heterozygous mice exhibit any morphological phenotypes? Also, for the hematoxylin analyses, were there any deficits in the conditional mutant mice within the CA regions of the hippocampus as well as the DG? Finally, It would be beneficial to quantify the fluorescence in panel F for SOX9, as was done in Figure 4.

3) Figure 7: A hemotoxylin analysis of the *Wnt3a cre* mutant strain would be very valuable here (similar to Figure. 1). Also, the data in Supplementary Figure 11 are very valuable, and I think they would be better served being included in this figure. Finally, the main phenotype the *Wnt3a cre* mice exhibit is a failure of the glial matrix arising from the fimbria to form. However, the formation and migration of TBR2 and PROX1 expressing cells appears relatively normal. As such, I think the Discussion of the manuscript need to be toned down to reflect this. For example, the first paragraph of the Discussion states that the fimbrial glial scaffold is necessary for the migration of granule neuron progenitors to the forming DG.

Revisions expected in follow-up work:

1) To demonstrate that fimbria glial scaffold defect is indeed the predominant cause of the phenotype, the authors need to provide addition data on (i) the fate of the ectopic cluster and (ii) smaller DG is not due to impaired adult neurogenesis.

a) What happens to those ectopic cluster of cells? The authors mentioned that those ectopic cells did not undergo cell death. Are they still there at 3 months old mice? Did they fail to migrate even the GFAP+ fimbria scaffold emerges after E18.5? If not, does this imply SOX9 controls some other important mechanisms that result in the reduced dentate gyrus phenotype? If so, the role of fimbria scaffold in brain morphogenesis may be overstated. Since the *Wnt3a-CreSox9* mutants die after E18.5, it is impossible to evaluate the contribution of *Sox9* loss in cortical hem to the reduced dentate gyrus phenotype. The authors may need to discuss on this issue.

b) The authors mentioned that the ectopic cells were absent in the adult brain without increased apoptosis. Again, when the glial scaffold emerged at P2 (Figure 4A-B), did the ectopic cells start to migrate so that the ectopic clusters disappeared later? Since *Sox9* are also expressed in the adult DG (Shin et al., 2015), it is possible that the smaller DG is due to loss of *Sox9* in the DG. The authors need to clarify this point, as the importance of the transient loss of fimbria glia scaffold may be exaggerated.

2) In Figure 4, the authors described that close to the DNE, the GFAP+ fimbria scaffold appears well separated from both DNE progenitors and those migrating in the 2ry matrix (Figure 4Fa). Along the migratory stream, progenitors start being more closely associated with the scaffold (Figure 4Fb). Finally, from the distal part of the fimbria, progenitors and scaffold appear completely intermingled (Figure 4Fc). From these, one can conclude that the fimbria scaffold is required for progenitor migration throughout the 2ry matrix. Indeed, the authors suggested in the model (Figure 8) that, progenitor migration towards the forming 3ry matrix relies on the fimbria scaffold. If so, one would expect cells that failed to migrate are found throughout the entire migration stream in the 2ry matrix. However, the ectopic cluster is only seen near the DNE. Does this imply only the delamination of progenitors is dependent on glia scaffold, rather than the continuous migration along the 2ry matrix? To answer this question, the authors could perform a sequential EdU/BrdU birthdate tracing of the cells found in the migration stream. If the migration really has defects, then one should be able to observe a different distribution of the birthdated cells in the migration stream. Also, in Figure 3Eii, the authors compared the % of TBR2+EdU+ cells in the ectopic cluster over total TBR2+Edu+ cells in 2ry matrix. This should be expanded to compare the % of such cells in the 1ry matrix, 2ry matrix, 3ry matrix over the total number of these cells in all matrices. This will provide much more information as to the migratory behavior of these cells.

---

## [Author Response]

Revisions for this paper:1) The primary finding of the manuscript is that SOX9-dependent fimbria glial scaffold guides the migration of granule neuron progenitors towards dentate gyrus. This is mainly supported by the phenotypic difference between *Sox9* knockout driven by *Sox1^Cre^* and *Nestin-Cr*e, with the former enables *Sox9* deletion at the cortical hem while the latter doesn't (Figure 1G). This makes proving the *Nestin-Cre* mutants have mild or no phenotypes particularly important. In several cases, the images actually suggest *Nestin-Cre Sox9* mutants do have a phenotype, but the statistical analysis indicate no significant difference (e.g. Figure 1A, the size of DG looks smaller but the graph indicates not; Figure 2G, no. of TBR2+ cells in the graph is reduced; Figure 4C, GFAP expression at E18.5 seems reduced in the graph). The authors should increase the sample size to provide more solid and consistent results. It would be useful to quantify GFAP fluorescence intensity in panel C, as was done in Figure 4.

We thank the reviewers and editors for their supportive comments. We agree that our conclusions rely on sometime delicate phenotypic differences. These are observed when *Sox9* is differentially deleted, in particular using *Sox1^Cre^* and *Nestin-Cre*, but also supported by studies of *Wnt3a^iresCre^* mutants. Some of the differences observed between *Sox1^Cre^* and *Nestin-Cre* mutant are clear, especially when examining the fimbrial part of the glial scaffold at E18.5 (Figure 4A), and correlate with their differential activity (Figure 1G-H). However, since both mutants show a disruption of DG development, other aspects of the phenotype, such as retention of cells in the ectopic cluster (Figure 2F) or secondary matrix (Figure 2M), appear more similar. Our interpretation, supported by exclusive deletion of *Sox9* in the CH, where *Nestin-Cre* is not active, is that the fimbrial scaffold offers a necessary, albeit transient, support for progenitor migration. Partial and variable *Nestin-Cre* activity in the CH might also explain the variability of migration defects observed in *Sox9^fl/fl^;Nestin-Cre* mutants. Indeed, loss of *Sox9* in DNE migrating cells may have a cell-autonomous effect. However, this would be a relatively subtle effect since we were not able to quantify any difference by examining progenitor numbers, proliferation and fate acquisition. The Discussion has been amended to reflect these elements.

To make the differences and similarities between *Sox1* and *Nestin-Cre* clearer, we have now increased the number of samples. We re-examined DG size quantification at P2 by adding 1 sample in controls and *Nestin-cre* mutants and 2 in *Sox9^fl/fl^;Sox1^Cre/+^*in Figure 1C. We still observe a significant reduction of DG size in *Sox1^Cre^* mutants, but not in *Nestin-Cre* ones.

We also added one P2 sample in both *Sox9^fl/fl^;Sox1^Cre/+^* and *Sox9^fl/fl^;Nestin-Cre* groups to quantify the total number of TBR2 and PROX1 cells at P2. While a reduction is observed in *Nestin-Cre* mutants, this still does not reach statistical significance (Figure 2G and J). We have amended the text to comment on this reduction.

Finally, we quantified GFAP fluorescence intensity in *Sox9^fl/fl^;Sox1^Cre/+^Sox9^fl/fl^;Nestin-Cre* and *Sox9^fl/fl^;Wnt3a^iresCre/+^*embryos at E18.5 and compared it to controls. We examined the fimbrial and supragranular scaffolds separately (Figure 6F). This analysis shows that expression of GFAP in the fimbrial scaffold is significantly reduced compared to controls in *Sox9^fl/fl^;Sox1^Cre/+^* and *Sox9^fl/fl^;Wnt3a^iresCre/+^*embryos, but not in *Nestin-Cre* ones. Conversely, GFAP expression in the supragranular scaffold is only significantly downregulated in *Sox9^fl/fl^;Sox1^Cre/+^* and *Sox9^fl/fl^;Nestin-Cre* embryos, but not in *Wnt3a^iresCre^* ones. This is consistent with our lineage tracing analyses demonstrating the fimbrial scaffold mostly derives from the CH while the supragranular part derives from the DNE. It is also consistent with a requirement for SOX9 in CH. A graph representing this analysis was added in Figure 6 (new panels E and F). We would like to thank the reviewers for this suggestion as this quantification further strengthens our results.

2) Figure 1: Do the *Sox9/Sox1 cre* heterozygous mice exhibit any morphological phenotypes? Also, for the hematoxylin analyses, were there any deficits in the conditional mutant mice within the CA regions of the hippocampus as well as the DG? Finally, It would be beneficial to quantify the fluorescence in panel F for SOX9, as was done in Figure 4.

We have now examined DG size in a *Sox9^fl/+^;Sox1^Cre/+^* adult mouse, and this does not appear to be different from controls (Figure 1—figure supplement 2). Therefore, in contrast with other organs where deletion of one copy of *Sox9* is detrimental, the DG appears normal.

We have also analysed CA regions in *Sox9^fl/fl^;Sox1^Cre/+^* mutants and compared to *Sox9^fl/+^;Sox1^+/+^* controls. We have found that while CA2 and CA3 layers looks slightly less compacted, the reduction in DG size is the most obvious defect in mutants. We have added this analysis to Figure 1—figure supplement 1.

In our original manuscript, we examined expression of SOX9/*Sox9* (Figure 1F, Figure 1—figure supplement 5A) and quantified expression of the gene by quantitative PCR (Figure 1—figure supplement 5B). These three independent analyses show the same result: SOX9/*Sox9* is not completely deleted in the archicortex in *Nestin-Cre* mutants, as previously shown (Winkler et al., 2018). The same *Nestin* enhancer has moreover been used to drive GFP expression in mice, and this again shows incomplete expression, although it should be borne in mind that this will be a different transgenic insertion and it may not be directly comparable (Li et al., 2009). Nevertheless, we believe that quantification of the SOX9 immunofluorescence, would not add any new element to our already clear and concordant results.

3) Figure 7: A hemotoxylin analysis of the *Wnt3a cre* mutant strain would be very valuable here (similar to Figure. 1). Also, the data in Supplementary Figure 11 are very valuable, and I think they would be better served being included in this figure. Finally, the main phenotype the *Wnt3a cre* mice exhibit is a failure of the glial matrix arising from the fimbria to form. However, the formation and migration of TBR2 and PROX1 expressing cells appears relatively normal. As such, I think the Discussion of the manuscript need to be toned down to reflect this. For example, the first paragraph of the Discussion states that the fimbrial glial scaffold is necessary for the migration of granule neuron progenitors to the forming DG.

We have performed a histological analysis of the developing DG in all 3 mutants at E18.5 and compared it to controls (Figure 1—figure supplement 3). While the overall morphology of the developing DG in *Sox9^fl/fl^;Wnt3a^iresCre/+^*embryos does not appear affected, we confirmed the presence of the ectopic cluster. We were unfortunately unable to generate sufficient numbers of samples to quantify the size of the DG in *Sox9^fl/fl^;Wnt3a^iresCre/+^*embryos.

As recommended, Supplementary Figure 11 has now been included in the main Figure 7.

*Sox9^fl/fl^;Wnt3a^iresCre/+^* present indeed a clear disruption of the fimbrial part of the scaffold. However, migration of TBR2+ intermediate progenitors is also affected since these cells are retained in the secondary matrix (Figure 7F). While this supports the requirement of the scaffold for migration, we understand that further analysis of the DG phenotype in these mutants is required to make the conclusion that it is required for DG morphogenesis. As recommended, we have toned down the first paragraph of the Discussion.

Revisions expected in follow-up work:1) To demonstrate that fimbria glial scaffold defect is indeed the predominant cause of the phenotype, the authors need to provide addition data on (i) the fate of the ectopic cluster and (ii) smaller DG is not due to impaired adult neurogenesis.

We agree that it would be very interesting to determine the fate of the ectopic cluster once the glial scaffold is recovered, more precisely understand if the cells are able to resume migration, or if retention in this ectopic location has permanently compromised their potential. Attempts so far have simply enabled us to report absence of the cluster in the adult, as visible in Figure 1Aii. Determining timing and modalities of its disappearance will require further investigations which are beyond the scope of this study, as acknowledged by the reviewers and editor. However, we believe that our current results allow us to conclude that adult neurogenesis does not play an exclusive role in mutant DG size reduction, for two reasons. First, we observe a smaller DG in very young animals, at P2, when the NSCs niche of the SGZ is not yet established. Secondly, the presence of a normal DG when using *Nestin-Cre* (Figure 1A-C), which is known to be expressed in SGZ NSCs, further argues that *Sox9* deletion in these cells does not result in major defects in DG morphology. As mentioned in the Discussion, it is nevertheless very likely that *Sox9* deletion affects adult neurogenesis, as we have observed this previously in the SVZ, where the protein is required for NSC maintenance. Conditional deletion of *Sox9* in SGZ NSCs is now required to determine its specific role during adult DG neurogenesis.

a) What happens to those ectopic cluster of cells? The authors mentioned that those ectopic cells did not undergo cell death. Are they still there at 3 months old mice? Did they fail to migrate even the GFAP+ fimbria scaffold emerges after E18.5? If not, does this imply SOX9 controls some other important mechanisms that result in the reduced dentate gyrus phenotype? If so, the role of fimbria scaffold in brain morphogenesis may be overstated. Since the *Wnt3a-Cre Sox9* mutants die after E18.5, it is impossible to evaluate the contribution of Sox9 loss in cortical hem to the reduced dentate gyrus phenotype. The authors may need to discuss on this issue.

We did not observe apoptosis in the ectopic cluster, but we only tested this at P2 (Figure 1—figure supplement 5); examinations at further stages are now required since the cluster is clearly absent in the adult (Figure 1Aii). As mentioned in the previous comment, it would be of interest to determine how it disappears and what happens to the cells. However, failure to migrate after retention in the ectopic cluster on the recovered scaffold may not only reflect a cell autonomous requirement for SOX9. The delay may permanently compromise the potential of the cells; for example, a signal, or sensitivity to it, could exist at E18.5 to promote migration, but not anymore at P2.

The loss of *Wnt3a^iresCre^* mutants indeed prevents assessment of the post-natal DG phenotype. In these mutants, the retention of TBR2 progenitors in the secondary matrix (Figure 7F), the presence of the ectopic cluster (Figure 7D, G, H) and the absence of the fimbrial scaffold (Figure 7I), as mostly observed in *Sox1^Cre^* mutants, all strongly support the role of the CH, where the gene is exclusively deleted, while the affected cells (ectopic cluster, TBR2 migrating cells) all retain *Sox9* expression. However, we agree that while all this evidence is strongly supportive, it does not formally demonstrate that reduction of adult DG size is exclusively due to absence of the fimbrial scaffold. We have thus toned down our Discussion as detailed above.

b) The authors mentioned that the ectopic cells were absent in the adult brain without increased apoptosis. Again, when the glial scaffold emerged at P2 (Figure 4A-B), did the ectopic cells start to migrate so that the ectopic clusters disappeared later? Since *Sox9* are also expressed in the adult DG (Shin et al., 2015), it is possible that the smaller DG is due to loss of *Sox9* in the DG. The authors need to clarify this point, as the importance of the transient loss of fimbria glia scaffold may be exaggerated.

The cluster is still present at P2, despite the recovery of the glial scaffold. Furthermore, the DG is already smaller at this stage, demonstrating that the loss of *Sox9* at or, as we show, before P2 results in reduction of the DG. We did not examine migration at later stages in the mutants, which would be of interest. However, as mentioned above, presence of a normal DG in *Nestin-Cre* mutants strongly suggest that absence of *Sox9* post-natally in DG does not result in reduction of its size.

2) In Figure 4, the authors described that close to the DNE, the GFAP+ fimbria scaffold appears well separated from both DNE progenitors and those migrating in the 2ry matrix (Figure 4Fa). Along the migratory stream, progenitors start being more closely associated with the scaffold (Figure 4Fb). Finally, from the distal part of the fimbria, progenitors and scaffold appear completely intermingled (Figure 4Fc). From these, one can conclude that the fimbria scaffold is required for progenitor migration throughout the 2ry matrix. Indeed, the authors suggested in the model (Figure 8) that, progenitor migration towards the forming 3ry matrix relies on the fimbria scaffold. If so, one would expect cells that failed to migrate are found throughout the entire migration stream in the 2ry matrix. However, the ectopic cluster is only seen near the DNE. Does this imply only the delamination of progenitors is dependent on glia scaffold, rather than the continuous migration along the 2ry matrix? To answer this question, the authors could perform a sequential EdU/BrdU birthdate tracing of the cells found in the migration stream. If the migration really has defects, then one should be able to observe a different distribution of the birthdated cells in the migration stream. Also, in Figure 3Eii, the authors compared the % of TBR2+EdU+ cells in the ectopic cluster over total TBR2+Edu+ cells in 2ry matrix. This should be expanded to compare the % of such cells in the 1ry matrix, 2ry matrix, 3ry matrix over the total number of these cells in all matrices. This will provide much more information as to the migratory behavior of these cells.

The 3D examination of progenitor migration along the scaffold provides interesting information and also further questions about the modalities of this process. We would like to thank the reviewer for these suggestions that would definitely be useful to better examine how the scaffold support progenitors and allow us to discriminate between defective delamination versus migration.